# Confidence Interval Estimation for the Common Mean of Several Zero-Inflated Gamma Distributions

**Theerapong Kaewprasert, Sa-Aat Niwitpong and Suparat Niwitpong ***

Department of Applied Statistics, Faculty of Applied Science, King Mongkut's University of Technology North Bangkok, Bangkok 10800, Thailand
* Correspondence: suparat.n@sci.kmutnb.ac.th

**Abstract:** In this study, we propose estimates for the confidence interval for the common mean of several zero-inflated gamma (ZIG) distributions based on the fiducial generalized confidence interval (GCI) and Bayesian and highest posterior density (HPD) methods based on the Jeffreys rule or uniform prior. Their performances in terms of their coverage probabilities and expected lengths are compared via a Monte Carlo simulation study. For almost all of the scenarios considered, the simulation results show that the fiducial GCI performed better than the Bayesian and HPD methods. Daily rainfall data from Chiang Mai Province, Thailand that contains several zero entries and follows a ZIG distribution is used to test the efficacies of the methods in real-world situations.

**Keywords:** zero-inflated gamma distribution; common mean; fiducial generalized confidence interval; Bayesian method

## 1. Introduction

Chiang Mai in northern Thailand is a region with high mountains and forests. When it comes to the rainy season, excess runoff water often floods villages at the foot of mountains. The rainy season in Thailand lasts from May to October. Between August and September, there is "heavy rain" to "very heavy rain", which can cause flash flooding and river bank overflows. However, predicting heavy precipitation and flooding is challenging due to rainfall variability. Daily rainfall data from the region typically include zero and positive values that fit a zero-inflated gamma (ZIG) distribution. Thus, to accurately predict future catastrophic events, it is vital to measure the central tendency of rainfall in specific places using statistical parameters such as the mean of a distribution. Therefore, the mean of a ZIG distribution can be used to analyze rainfall data series for forecasting future precipitation amounts.

The ZIG distribution can be used to fit data that contain both zero and positive values: the positive values follow a gamma distribution, while the zero values follow a binomial distribution. The ZIG distribution has been used to analyze right-skewed data with a high clump-at-zero frequency in several fields. In meteorology, Kaewprasert et al. [1] used the mean of a ZIG distribution to analyze mixed zero and non-zero rainfall data. In medicine, Wang et al. [2] analyzed data on the HIV status of children recorded as non-responses (zero) and positive responses by using the mean of a ZIG distribution.

The two types of statistical inference are parameter estimation and hypothesis testing. The most often used interval estimation technique for a parameter is the confidence interval enclosing the estimate's minimum and maximum values. The confidence interval for the mean of a ZIG distribution has been the focus of several studies. Muralidharan and Kale et al. [3] estimated the confidence interval for the mean of a ZIG distribution and applied it to analyze real rainfall data. Simultaneous confidence intervals for the difference between the means of ZIG distributions were introduced by Ren et al. [4].

Kaewprasert et al. [1] expanded the scope of this by comparing the difference between the means of several ZIG distributions. Wang et al. [2] produced confidence interval estimates for the mean of a gamma distribution with zero values.

The mean is frequently utilized in practice to gauge statistical significance in many domains. The common mean is of interest when establishing inference for more than one population when independent samples are collected from them. The process of building the confidence interval for the common mean of several distributions has been studied by many scholars. For example, Yan [5] established confidence interval estimation for the common mean of several gamma populations by using fiducial inference and the method of variance estimates recovery (MOVER). Maneerat and Niwitpong [6] estimated the confidence interval for the common mean of several delta-lognormal distributions. As previously indicated, although there have been numerous estimations of the confidence interval for the common mean of several gamma and delta-lognormal distributions, the common mean of several ZIG distributions has not yet been the subject of a study on statistical inference.

The motivation for the study was to examine previously reported confidence interval estimation methods for the common mean of several gamma distributions and extend them to estimate the confidence interval for the common mean of several ZIG distributions. Thus, we chose estimation methods used for the confidence interval for the common mean and common coefficient of variation of several delta-lognormal distributions as follows. Maneerat and Niwitpong [6] proposed using the fiducial generalized confidence interval (GCI) and the highest posterior density (HPD) interval based on the Jeffreys rule prior to estimate the confidence interval for the common mean of several delta-lognormal distributions. Using the fiducial GCI and Bayesian approach based on the uniform prior was proposed by Yosboonruang et al. [7] to estimate the confidence interval for the common coefficient of variation of several delta-lognormal distributions.

Herein, we explored several confidence interval estimation methods for the common mean of several ZIG distributions using the fiducial GCI approach and Bayesian and HPD methods based on the Jeffreys rule or uniform prior. We used them to calculate the 95% confidence interval for the common mean of three daily rainfall datasets (Chomthong, Mae Taeng, and Doi Saket) in Chiang Mai, Thailand.

The outline of this study is organized as follows. Section 2 provides the methodologies to estimate the confidence interval for the common mean of several ZIG distributions. Section 3 reports the numerical computations using the methods in a Monte Carlo simulation study. Section 4 presents the empirical application of the proposed confidence interval estimation methods using data on daily rainfall collected from three rain stations in Chiang Mai, Thailand, in September 2020 and 2021. Finally, a discussion and conclusions are offered in Sections 5 and 6, respectively.

## 2. Methods

Let $Y_{ij}$; $i = 1, 2, \ldots, k$; $j = 1, 2, \ldots, n_i$ be random variables of size $n_i$ from $k$ ZIG distributions denoted as $Y_{ij} \sim \text{ZIG}(a_i, b_i, \delta_i)$. This distribution has three parameters: shape parameter $a_i$, rate parameter $b_i$, and the proportion of zero values $\delta_i$. For $k$ populations of observations, the distribution function of $Y_{ij}$ is given by

$$F(y_{ij}; a_i, b_i, \delta_i) = \begin{cases} \delta_i & ; y_{ij} = 0 \\ \delta_i + (1 - \delta_i) G(y_{ij}; a_i, b_i) & ; y_{ij} > 0 \end{cases}, \tag{1}$$

where $G(y_{ij}; a_i, b_i)$ is a gamma distribution function, which can be denoted as $G_{ij}$ when $Y_{ij} \neq 0$; $i = 1, 2, \ldots, k$; $j = 1, 2, \ldots, n_{i(1)}$. For $Y_{ij} = 0$, the zero observations follow a binomial distribution denoted as $n_{i(0)} \sim \text{B}(n_i, \delta_i)$. Furthermore, $n_{i(0)}$ and $n_{i(1)}$ represent the numbers of zero and non-zero values, respectively, where $n_{i(0)} + n_{i(1)} = n_i$. The population mean of $Y_{ij}$ is given by

$$\eta_i = (1 - \delta_i) \frac{a_i}{b_i}. \tag{2}$$

Krishnamoorthy et al. [8] and Krishnamoorthy and Wang [9] used cube-root approximation $X_{ij} = G_{ij}^{1/3}$, thereby ensuring that the $X'_{ij}s$ are approximately normally distributed, which is denoted as $X_{ij} \sim \mathrm{N}(\mu_i, \sigma_i^2)$ with mean and variance of $\mu_i$ and $\sigma_i^2$, respectively. It is possible to represent $\mu_i$ and $\sigma_i^2$ in terms of $a_i$ and $b_i$, respectively, as follows:

$$\mu_i = \left(\frac{a_i}{b_i}\right)^{1/3}\left(1 - \frac{1}{9a_i}\right) \qquad \text{and} \qquad \sigma_i^2 = \frac{1}{9a_i^{1/3}b_i^{2/3}}. \tag{3}$$

By resolving the equations in the $a_i$ and $b_i$ sets above, we, respectively, arrive at

$$a_i = \frac{1}{9}\left\{\left(1 + \frac{\mu_i^2}{2\sigma_i^2}\right) + \left[\left(1 + \frac{\mu_i^2}{2\sigma_i^2}\right)^2 - 1\right]^{1/2}\right\} \qquad \text{and} \qquad b_i = \frac{1}{27a_i^{1/2}(\sigma_i^2)^{3/2}}. \tag{4}$$

Thus, the mean of a ZIG distribution is $\eta_i = (1 - \delta_i)\frac{a_i}{b_i} = (1 - \delta_i)\left(\frac{\mu_i}{2} + \sqrt{\frac{\mu_i^2}{4} + \sigma_i^2}\right)^3$.

The unbiased estimators for $\mu_i$, $\sigma_i^2$, and $\delta_i$ are $\hat{\mu}_i = \frac{1}{n_{i(1)}}\sum_{j=1}^{n_{i(1)}} y_{ij}^{1/3}$, $\hat{\sigma}_i^2 = \frac{1}{n_{i(1)}-1}$ $\sum_{j=1}^{n_{i(1)}}\left(y_{ij}^{1/3} - \hat{\mu}_i\right)^2$, and $\hat{\delta}_i = n_{i(0)}/n_i$, respectively; then

$$\hat{\eta}_i = (1 - \hat{\delta}_i)\frac{\hat{a}_i}{\hat{b}_i} = (1 - \hat{\delta}_i)\left(\frac{\hat{\mu}_i}{2} + \sqrt{\frac{\hat{\mu}_i^2}{4} + \hat{\sigma}_i^2}\right)^3, \tag{5}$$

where $\hat{a}_i = \frac{1}{9}\left\{\left(1 + \frac{\hat{\mu}_i^2}{2\hat{\sigma}_i^2}\right) + \left[\left(1 + \frac{\hat{\mu}_i^2}{2\hat{\sigma}_i^2}\right)^2 - 1\right]^{1/2}\right\}$ and $\hat{b}_i = \frac{1}{27\hat{a}_i^{1/2}(\hat{\sigma}_i^2)^{3/2}}$.

Using the finding from Aitchison [10], Vännman [11] claimed that the minimal variance unbiased estimators of the variance of $\hat{\eta}_i$ can be derived as

$$\hat{V}(\hat{\eta}_i) \approx \frac{n_i(\hat{a}_i + 1) - \hat{a}_i n_{i(1)}}{(n_i - 1)(\hat{a}_i n_{i(1)} + 1)}\left((1 - \hat{\delta}_i)\frac{\hat{a}_i}{\hat{b}_i}\right)^2. \tag{6}$$

According to Yan [5] and Maneerat and Niwitpong [6], the common mean of several ZIG distributions can be defined as

$$\tilde{\eta} = \frac{\sum_{i=1}^k w_i \hat{\eta}_i}{\sum_{i=1}^k w_i}, \tag{7}$$

where $w_i = 1/\hat{V}(\hat{\eta}_i)$. The confidence interval for the common mean of several ZIG distributions can be estimated by using the suggested methods listed below.

*2.1. The Fiducial GCI Method*

Fisher [12] was the first to propose the fiducial approach. Meanwhile, Hannig [13] conducted additional research into the fiducial approach and provided some general results. The fiducial interval is the generalized pivotal quantity (GPQ), which may be applied in generalized inference and can be viewed as the result of the fiducial framework. A framework for this that shows the connection between the distribution and the parameter was proposed by Hannig et al. [14] in the form of a fiducial GPQ.

Suppose $Y_{ij}$; $i = 1, 2, \ldots, k$; $j = 1, 2, \ldots, n_i$ is a random sample from $\mathrm{ZIG}(a_i, b_i, \delta_i)$, where $\tau_i = (a_i, b_i, \delta_i)$ is the parameter of interest. Therefore, the GPQ $T(Y_{ij}; y_{ij}, a_i, b_i, \delta_i)$ can only be a function of $\tau_i$. This is called the fiducial GPQ, which satisfies the following two conditions:

1. For each $y_{ij}$, the conditional distribution of $T(Y_{ij}; y_{ij}, a_i, b_i, \delta_i)$ is free of the nuisance parameter.

2.  For the observed value of $T(Y_{ij}; y_{ij}, a_i, b_i, \delta_i)$ at $Y_{ij} = y_{ij}$, $t(y_{ij}; y_{ij}, a_i, b_i, \delta_i) = \tau_i$.

According to Krishnamoorthy et al. [8], this approach is based on the observation that $X_{ij} \sim N(\mu_i, \sigma_i^2)$ approximates a gamma distribution. Let $\hat{\mu}_i$ and $\hat{\sigma}_i^2$, respectively, represent the observed sample mean and variance based on the $X_{ij}'s$ that have been cube-root transformed. This makes it possible to obtain the respective fiducial GPQs for $\mu_i$ and $\sigma_i^2$ as follows:

$$T_{\mu_i} = \hat{\mu}_i + \frac{U_i}{\sqrt{V_i}} \sqrt{\frac{\left(n_{i(1)} - 1\right)\hat{\sigma}_i^2}{n_{i(1)}}} \qquad \text{and} \qquad T_{\sigma_i^2} = \frac{\left(n_{i(1)} - 1\right)\hat{\sigma}_i^2}{V_i}, \tag{8}$$

where $\hat{\mu}_i = \frac{1}{n_{i(1)}} \sum_{j=1}^{n_{i(1)}} y_{ij}^{1/3}$, $\hat{\sigma}_i^2 = \frac{1}{n_{i(1)}-1} \sum_{j=1}^{n_{i(1)}} \left(y_{ij}^{1/3} - \hat{\mu}_i\right)^2$, $U_i \sim N(0,1)$, and $V_i \sim \chi_{n_{i(1)}-1}^2$. In addition, the respective fiducial GPQs for $a_i$ and $b_i$ have the following forms:

$$T_{a_i} = \frac{1}{9}\left\{ \left(1 + \frac{T_{\mu_i}^2}{2T_{\sigma_i^2}}\right) + \left[\left(1 + \frac{T_{\mu_i}^2}{2T_{\sigma_i^2}}\right)^2 - 1\right]^{1/2} \right\} \qquad \text{and} \qquad T_{b_i} = \frac{1}{27 T_{a_i}^{1/2} T_{\sigma_i^2}^{3/2}}. \tag{9}$$

Similarly, the fiducial GPQ for $\delta_i$ can be written as [15]

$$T_{\delta_i} \sim \frac{1}{2} Beta\left(n_{i(0)} + 1, n_{i(1)}\right) + \frac{1}{2} Beta\left(n_{i(0)}, n_{i(1)} + 1\right). \tag{10}$$

Meanwhile, the fiducial GPQ for $\hat{\eta}_i$ is given by

$$T_{\hat{\eta}_i} = (1 - T_{\delta_i}) \frac{T_{a_i}}{T_{b_i}} = (1 - T_{\delta_i})\left(\frac{T_{\mu_i}}{2} + \sqrt{\frac{T_{\mu_i}^2}{4} + T_{\sigma_i^2}}\right)^3. \tag{11}$$

Subsequently, the fiducial GPQ for the estimated variance of $\hat{\eta}_i$ is given by

$$T_{\hat{V}(\hat{\eta}_i)} = \frac{n_i(T_{a_i} + 1) - T_{a_i} n_{i(1)}}{(n_i - 1)(T_{a_i} n_{i(1)} + 1)}\left((1 - T_{\delta_i})\frac{T_{a_i}}{T_{b_i}}\right)^2. \tag{12}$$

Therefore, we can estimate the confidence interval for the common mean of $k$ ZIG distributions ($\tilde{\eta}$) using its fiducial GPQ as follows:

$$T_{\tilde{\eta}} = \frac{\sum_{i=1}^{k} T_{w_i} T_{\hat{\eta}_i}}{\sum_{i=1}^{k} T_{w_i}}, \tag{13}$$

where $T_{w_i} = 1/T_{\hat{V}(\hat{\eta}_i)}$.

Thus, the $100(1 - \gamma)\%$ fiducial GCI for $\tilde{\eta}$ becomes

$$CI_{\tilde{\eta}}^{FGCI} = [L_{\tilde{\eta}}^{FGCI}, U_{\tilde{\eta}}^{FGCI}] = [T_{\tilde{\eta}}(\gamma/2), T_{\tilde{\eta}}(1 - \gamma/2)]. \tag{14}$$

where $T_{\tilde{\eta}}(\gamma)$ denotes the $\gamma^{th}$ percentiles of $T_{\tilde{\eta}}$. This process is specified in Algorithm 1.

---

**Algorithm 1** The fiducial GCI method

---

1.  Generate $Y_{ij} \sim ZIG(a_i, b_i, \delta_i)$; $i = 1, 2, \ldots, k$; $j = 1, 2, \ldots, n_i$.
2.  Generate $U_i \sim N(0, 1)$ and $V_i \sim \chi_{n_{i(1)}-1}^2$ independently.
3.  Compute fiducial GPQs $T_{a_i}$, $T_{b_i}$, and $T_{\delta_i}$.
4.  Compute $T_{\hat{\eta}_i}$ and $T_{\hat{V}(\hat{\eta}_i)}$, leading to obtaining $T_{\tilde{\eta}}$.
5.  Repeat steps (1)–(4) 2000 times.
6.  Compute the 95% fiducialGCI for $T_{\tilde{\eta}}$ using Equation (14).

---

*2.2. The Bayesian Methods*

Suppose that $Y_{ij} \neq 0$; $i = 1, 2, \ldots, k$; $j = 1, 2, \ldots, n_{i(1)}$, then $Y_{ij}^{1/3} = X_{ij} \sim N(\mu_i, \sigma_i^2)$. Likelihood function $p(x_{ij} \mid \mu_i, \sigma_i^2)$ for $x_{ij}$ and the prior distribution, which is used to explain conditional probability, make up the Bayesian statistical approach. Therefore, the likelihood function of $k$ normally distributed samples is given by

$$p(x_{ij} \mid \mu_i, \sigma_i^2) \propto \prod_{i=1}^{k} (\sigma_i^2)^{-\frac{n_{i(1)}}{2}} \exp\left[ -\frac{1}{2\sigma_i^2} \sum_{j=1}^{n_{i(1)}} (x_{ij} - \mu_i)^2 \right]. \tag{15}$$

For the ZIG distribution, the joint likelihood function of $k$ individual samples is given by

$$p(y_{ij} \mid \mu_i, \sigma_i^2, \delta_i) \propto \prod_{i=1}^{k} \delta_i^{n_{i(0)}} (1 - \delta_i)^{n_{i(1)}} (\sigma_i^2)^{-\frac{n_{i(1)}}{2}} \exp\left[ -\frac{1}{2\sigma_i^2} \sum_{j=1}^{n_{i(1)}} \left( y_{ij}^{1/3} - \mu_i \right)^2 \right]. \tag{16}$$

The common mean for several ZIG distributions can be estimated using the Bayesian approach based on a variety of priors, two of which are derived in the following subsections.

2.2.1. The Jeffreys Rule Prior

Introduced by Harvey and Van Der Merwe [16], the Jeffreys rule prior can be written as

$$p(\tilde{\eta})_{Baye.J} \propto \prod_{i=1}^{k} \frac{1}{\sigma_i^3} \delta_i^{-1/2} (1 - \delta_i)^{1/2}. \tag{17}$$

Adding the likelihood functions in Equations (16) and (17) results in the posterior distribution of $\eta$ becoming

$$
\begin{aligned}
p(\tilde{\eta} \mid y_{ij})_{Baye.J} \quad &\propto \prod_{i=1}^{k} \delta_i^{n_{i(0)} - 1/2} (1 - \delta_i)^{n_{i(1)} + 1/2} (\sigma_i^2)^{-\frac{n_{i(1)} + 3}{2}} \exp\left[ -\frac{1}{2\sigma_i^2} \sum_{j=1}^{n_{i(1)}} \left( y_{ij}^{1/3} - \mu_i \right)^2 \right] \\
&\propto \prod_{i=1}^{k} \delta_i^{(n_{i(0)} + 1/2) - 1} (1 - \delta_i)^{(n_{i(1)} + 3/2) - 1} \frac{\sqrt{n_{i(1)}}}{\sqrt{2\pi\sigma_i^2}} \exp\left( -\frac{n_{i(1)}}{2\sigma_i^2} (\mu_i - \hat{\mu}_i)^2 \right) \\
&\times \frac{\left( \frac{(n_{i(1)} - 1)\hat{\sigma}_i^2}{2} \right)^{\frac{n_{i(1)} + 1}{2}}}{\Gamma\left( \frac{n_{i(1)} + 1}{2} \right)} (\sigma_i^2)^{-\frac{n_{i(1)} + 1}{2} - 1} \exp\left( -\frac{(n_{i(1)} - 1)\hat{\sigma}_i^2}{2\sigma_i^2} \right),
\end{aligned}
\tag{18}
$$

where $\hat{\mu}_i = \frac{1}{n_{i(1)}} \sum_{j=1}^{n_{i(1)}} y_{ij}^{1/3}$ and $\hat{\sigma}_i^2 = \frac{1}{n_{i(1)} - 1} \sum_{j=1}^{n_{i(1)}} \left( y_{ij}^{1/3} - \hat{\mu}_i \right)^2$. Subsequently, the respective marginal posterior distributions of $\mu_i$, $\sigma_i^2$, and $\delta_i$ are obtained as

$$
\begin{aligned}
\mu_i^{(Baye.J)} &\mid \sigma_i^2, y_{ij} \sim N(\hat{\mu}_i, \sigma_i^2/n_{i(1)}) \\
\sigma_i^{2(Baye.J)} &\mid y_{ij} \sim IG((n_{i(1)} + 1)/2, (n_{i(1)} - 1)\hat{\sigma}_i^2/2) \\
\delta_i^{(Baye.J)} &\mid y_{ij} \sim Beta(n_{i(0)} + 1/2, n_{i(1)} + 3/2).
\end{aligned}
\tag{19}
$$

In addition, the respective Bayesian derivations for $a_i$ and $b_i$ based on the Jeffreys rule prior have the following forms:

$$a_i^{(Baye.J)} = \frac{1}{9} \left\{ \left( 1 + \frac{(\mu_i^{(Baye.J)})^2}{2\sigma_i^{2(Baye.J)}} \right) + \left[ \left( 1 + \frac{(\mu_i^{(Baye.J)})^2}{2\sigma_i^{2(Baye.J)}} \right)^2 - 1 \right]^{1/2} \right\} \tag{20}$$

and

$$b_i^{(Baye.J)} = \frac{1}{27 \left( a_i^{(Baye.J)} \right)^{1/2} \left( \sigma_i^{2(Baye.J)} \right)^{3/2}}. \tag{21}$$

Thus, the Bayesian estimation for $\hat{\eta}_i$ based on the Jeffreys rule prior is given by

$$\hat{\eta}_i^{(Baye.J)} = (1 - \delta_i^{(Baye.J)}) \frac{a_i^{(Baye.J)}}{b_i^{(Baye.J)}}$$

$$= (1 - \delta_i^{(Baye.J)}) \left( \frac{\mu_i^{(Baye.J)}}{2} + \sqrt{\frac{(\mu_i^{(Baye.J)})^2}{4} + \sigma_i^{2(Baye.J)}} \right)^3 . \tag{22}$$

Meanwhile, the Bayesian estimation for the variance of $\hat{\eta}_i$ based on the Jeffreys rule prior is given by

$$\hat{V}(\hat{\eta}_i)^{(Baye.J)} = \frac{n_i(a_i^{(Baye.J)} + 1) - a_i^{(Baye.J)} n_{i(1)}}{(n_i - 1)(a_i^{(Baye.J)} n_{i(1)} + 1)} \left( (1 - \delta_i^{(Baye.J)}) \frac{a_i^{(Baye.J)}}{b_i^{(Baye.J)}} \right)^2 . \tag{23}$$

Therefore, we can construct the Bayesian credible interval for the common mean of several ZIG distributions based on the Jeffreys rule prior as

$$\tilde{\eta}^{(Baye.J)} = \frac{\sum_{i=1}^k w_i^{(Baye.J)} \hat{\eta}_i^{(Baye.J)}}{\sum_{i=1}^k w_i^{(Baye.J)}}, \tag{24}$$

where $w_i^{(Baye.J)} = 1/\hat{V}(\hat{\eta}_i)^{(Baye.J)}$.

Thus, the $100(1 - \gamma)\%$ Bayesian credible interval for $\tilde{\eta}$ based on the Jeffreys rule prior is

$$CI_{\tilde{\eta}}^{Baye.J} = [L_{\tilde{\eta}}^{Baye.J}, U_{\tilde{\eta}}^{Baye.J}] = [\tilde{\eta}^{(Baye.J)}(\gamma/2), \tilde{\eta}^{(Baye.J)}(1 - \gamma/2)]. \tag{25}$$

where $\tilde{\eta}^{(Baye.J)}(\gamma)$ denotes the $\gamma^{th}$ percentiles of $\tilde{\eta}^{(Baye.J)}$.

2.2.2. The Uniform Prior

Due to the uniform prior's constant function for the prior probability, Bolstad and Curran [17] presented the uniform priors of $p(\mu_i)_{Baye.U} \propto 1$, $p(\sigma_i^2)_{Baye.U} \propto 1$, and $p(\delta_i)_{Baye.U} \propto 1$. Subsequently, the posterior distribution of $\tilde{\eta}$ based on the uniform prior becomes

$$p(\tilde{\eta} \mid y_{ij})_{Baye.U} \propto \prod_{i=1}^k \delta_i^{n_{i(0)}} (1 - \delta_i)^{n_{i(1)}} (\sigma_i^2)^{-\frac{n_{i(1)}}{2}} \exp\left[ -\frac{1}{2\sigma_i^2} \sum_{j=1}^{n_{i(1)}} \left( y_{ij}^{1/3} - \mu_i \right)^2 \right]$$

$$\propto \prod_{i=1}^k \delta_i^{(n_{i(0)}+1)-1} (1 - \delta_i)^{(n_{i(1)}+1)-1} \frac{\sqrt{n_{i(1)}}}{\sqrt{2\pi\sigma_i^2}} \exp\left( -\frac{n_{i(1)}}{2\sigma_i^2} (\mu_i - \hat{\mu}_i)^2 \right)$$

$$\times \frac{\left( \frac{(n_{i(1)}-1)\hat{\sigma}_i^2}{2} \right)^{\frac{n_{i(1)}-2}{2}}}{\Gamma\left( \frac{n_{i(1)}-2}{2} \right)} (\sigma_i^2)^{-\frac{n_{i(1)}-2}{2}-1} \exp\left( -\frac{(n_{i(1)}-1)\hat{\sigma}_i^2}{2\sigma i^2} \right), \tag{26}$$

where $\hat{\mu}_i = \frac{1}{n_{i(1)}} \sum_{j=1}^{n_{i(1)}} y_{ij}^{1/3}$ and $\hat{\sigma}_i^2 = \frac{1}{n_{i(1)}-1} \sum_{j=1}^{n_{i(1)}} \left( y_{ij}^{1/3} - \hat{\mu}_i \right)^2$. Consequentially, the respective marginal posterior distributions of $\mu_i$, $\sigma_i^2$, and $\delta_i$ can be obtained as

$$\mu_i^{(Baye.U)} \mid \sigma_i^2, y_{ij} \sim N(\hat{\mu}_i, \sigma_i^2/n_{i(1)})$$
$$\sigma_i^{2(Baye.U)} \mid y_{ij} \sim IG((n_{i(1)} - 2)/2, (n_{i(1)} - 1)\hat{\sigma}_i^2/2) \tag{27}$$
$$\delta_i^{(Baye.U)} \mid y_{ij} \sim Beta(n_{i(0)} + 1, n_{i(1)} + 1).$$

In addition, the Bayesian uniform priors for $a_i$ and $b_i$ have the following respective forms:

$$a_i^{(Baye.U)} = \frac{1}{9}\left\{ \left(1 + \frac{(\mu_i^{(Baye.U)})^2}{2\sigma_i^{2(Baye.U)}}\right) + \left[\left(1 + \frac{(\mu_i^{(Baye.U)})^2}{2\sigma_i^{2(Baye.U)}}\right)^2 - 1\right]^{1/2}\right\} \quad (28)$$

and

$$b_i^{(Baye.U)} = \frac{1}{27\left(a_i^{(Baye.U)}\right)^{1/2}\left(\sigma_i^{2(Baye.U)}\right)^{3/2}}. \quad (29)$$

Thus, the Bayesian estimate for $\hat{\eta}_i$ based on the uniform prior is given by

$$\begin{aligned}
\hat{\eta}_i^{(Baye.U)} &= (1 - \delta_i^{(Baye.U)})\frac{a_i^{(Baye.U)}}{b_i^{(Baye.U)}} \\
&= (1 - \delta_i^{(Baye.U)})\left(\frac{\mu_i^{(Baye.U)}}{2} + \sqrt{\frac{(\mu_i^{(Baye.U)})^2}{4} + \sigma_i^{2(Baye.U)}}\right)^3.
\end{aligned} \quad (30)$$

Subsequently, the Bayesian estimate for the variance of $\hat{\eta}_i$ based on the uniform prior is given by

$$\hat{V}(\hat{\eta}_i)^{(Baye.U)} = \frac{n_i(a_i^{(Baye.U)} + 1) - a_i^{(Baye.U)}n_{i(1)}}{(n_i - 1)(a_i^{(Baye.U)}n_{i(1)} + 1)}\left((1 - \delta_i^{(Baye.U)})\frac{a_i^{(Baye.U)}}{b_i^{(Baye.U)}}\right)^2. \quad (31)$$

Therefore, we can construct the Bayesian estimate for the confidence interval for the common mean of several ZIG distributions based on the uniform prior as

$$\tilde{\eta}^{(Baye.U)} = \frac{\sum_{i=1}^k w_i^{(Baye.U)}\hat{\eta}_i^{(Baye.U)}}{\sum_{i=1}^k w_i^{(Baye.U)}}, \quad (32)$$

where $w_i^{(Baye.U)} = 1/\hat{V}(\hat{\eta}_i)^{(Baye.U)}$.

Thus, the $100(1 - \gamma)\%$ Bayesian credible interval for $\tilde{\eta}$ based on the uniform prior can be written as

$$CI_{\tilde{\eta}}^{Baye.U} = [L_{\tilde{\eta}}^{Baye.U}, U_{\tilde{\eta}}^{Baye.U}] = [\tilde{\eta}^{(Baye.U)}(\gamma/2), \tilde{\eta}^{(Baye.U)}(1 - \gamma/2)]. \quad (33)$$

where $\tilde{\eta}^{(Baye.U)}(\gamma)$ denotes the $\gamma$th percentiles of $\tilde{\eta}^{(Baye.U)}$.

### 2.3. The HPD Interval

In the previous section, the Bayesian statistical approach is made up of the prior distribution, which is used to define the conditional probability, and likelihood function $p(y_{ij} \mid \lambda)$, where $\lambda = (\mu_i, \sigma_i^2, \delta_i)$. Therefore, the posterior distribution of $\lambda$ is given by

$$p(\lambda \mid y_{ij}) \propto p(\lambda)p(y_{ij} \mid \lambda). \quad (34)$$

When posterior distribution $p(\lambda \mid y_{ij})$ is not symmetric, Box and Tiao [18] introduced the HPD interval with the characteristic that the probability density of each point inside the interval is greater than that of every point outside of it. Consequently, region $W$ in the parameter space of $\lambda$ is known as the HPD region of the content $(1 - \gamma)$. These are the two conditions that comprise this situation:

1. $\Pr(\lambda \in W \mid y_{ij}) = 1 - \gamma$.
2. For $\lambda_1 \in W$ and $\lambda_2 \notin W$, $p(\lambda_1 \mid y_{ij}) \geq p(\lambda_2 \mid y_{ij})$.

Similar to the studies of Maneerat and Niwitpong [6], Yosboonruang et al. [7], Chen and Shao [19], and Noyan and Pham-Gia [20], we applied the *HPDinterval* package in the R software suite for Step (6) in Algorithm 2 to respectively compute the HPD intervals based on the Jeffreys rule or uniform prior for $\tilde{\eta}$ as follows:

$$CI_{\tilde{\eta}}^{HPD.J} = [L_{\tilde{\eta}}^{HPD.J}, U_{\tilde{\eta}}^{HPD.J}] = [\tilde{\eta}^{(HPD.J)}(\gamma/2), \tilde{\eta}^{(HPD.J)}(1-\gamma/2)] \tag{35}$$

and

$$CI_{\tilde{\eta}}^{HPD.U} = [L_{\tilde{\eta}}^{HPD.U}, U_{\tilde{\eta}}^{HPD.U}] = [\tilde{\eta}^{(HPD.U)}(\gamma/2), \tilde{\eta}^{(HPD.U)}(1-\gamma/2)]. \tag{36}$$

---

**Algorithm 2** The Bayesian credible interval base on the Jeffreys rule or uniform prior

---

1. Generate $Y_{ij} \sim \text{ZIG}(a_i, b_i, \delta_i); i = 1, 2, \ldots, k; j = 1, 2, \ldots, n_i$.
2. Compute $\hat{\mu}_i$ and $\hat{\sigma}_i^2$.
3. Generate $\mu_i^{(Baye.J)}$, $\sigma_i^{2(Baye.J)}$, and $\delta_i^{(Baye.J)}$ as given in Equation (19) and $\mu_i^{(Baye.U)}$, $\sigma_i^{2(Baye.U)}$, and $\delta_i^{(Baye.U)}$ as given in Equation (27) based on the Jeffreys rule or uniform prior, respectively.
4. Compute $\hat{\eta}_i^{(Baye.J)}$ and $\hat{V}(\hat{\eta}_i)^{(Baye.J)}$ to obtain $\tilde{\eta}^{(Baye.J)}$ as given in Equation (24), and $\hat{\eta}_i^{(Baye.U)}$ and $\hat{V}(\hat{\eta}_i)^{(Baye.U)}$ to obtain $\tilde{\eta}^{(Baye.U)}$ as given in Equation (32), respectively.
5. Repeat steps (1)–(4) 2000 times.
6. Compute the 95% Bayesian credible interval based on the Jeffreys rule or uniform prior for $\tilde{\eta}$ as given in Equations (25) and (33), respectively.

---

## 3. The Monte Carlo Simulation Study and Results

### 3.1. Simulation Results

This was conducted using the R statistical program to investigate the effectiveness of the estimation methods for the confidence interval for the common mean of several ZIG distributions. The metrics used for the comparison are the coverage probability (CP), which is the percentage of times that the real parameter value is contained within the confidence interval for $\tilde{\eta}$; lower and upper error probabilities (LEP and UEP, respectively); and the expected length (EL), which is the average length of the confidence interval for $\tilde{\eta}$. The confidence interval estimation method that performs best for a particular scenario is the one with a coverage probability close to or greater than the nominal confidence level of 0.95 and the shortest expected length, while the required values of LEP and UEP are balanced at 0.025. The number of generated random samples was fixed at 10,000 replications with 2000 pivotal quantities for the fiducial GCI, the Bayesian, and HPD methods. We set the sample sizes ($n_i$), the proportion of zero values ($\delta_i$), and the shape parameter ($a_i$) as reported in Tables 1 and 2 for $k = 3$ and $k = 5$, respectively. Finally, rate parameter ($b_i$) was set as 1.0. In this study, the criterion to compare the efficiencies of the confidence intervals (CIs) are CPs and ELs, where CP is the percentage of time that the true parameter value is contained within the interval, and EL is the average length of the CIs. First, the confidence intervals were considered by the CPs. Since the nominal confidence level was 0.95, then the CIs which provided CPs equal to or more than 0.95 are selected. After that, the ELs of these CIs are considered to find the shortest length to be the best CI.

**Table 1.** The coverage probabilities and expected lengths for estimating the 95% confidence interval for the common mean of several ZIG distributions ($k = 3$).

| $n_i$ | $\delta_i$ | $a_i$ | FGCI | | | Baye.J | | | Baye.U | | | HPD.J | | | HPD.U | | |
|---|---|---|---|---|---|---|---|---|---|---|---|---|---|---|---|---|---|
| | | | LEP | CP (EL) | UEP | LEP | CP (EL) | UEP | LEP | CP (EL) | UEP | LEP | CP (EL) | UEP | LEP | CP (EL) | UEP |
| $30_3$ | $0.2_3$ | $12.0_3$ | 0.0000 | 0.9399 (2.3165) | 0.0601 | 0.0000 | 0.9565 (3.1680) | 0.0435 | 0.0000 | 0.9363 (3.2583) | 0.0637 | 0.0001 | 0.9689 (3.0892) | 0.0310 | 0.0000 | 0.9503 (3.1848) | 0.0497 |
| | | $12.5_3$ | 0.0000 | 0.9486 (2.4154) | 0.0514 | 0.0000 | 0.9653 (3.3053) | 0.0347 | 0.0000 | 0.9478 (3.3994) | 0.0522 | 0.0002 | 0.9754 (3.2230) | 0.0244 | 0.0000 | 0.9616 (3.3230) | 0.0384 |
| | | $13.0_3$ | 0.0000 | 0.9503 (2.5156) | 0.0497 | 0.0000 | 0.9674 (3.4421) | 0.0326 | 0.0000 | 0.9513 (3.5403) | 0.0487 | 0.0001 | 0.9763 (3.3563) | 0.0236 | 0.0000 | 0.9640 (3.4604) | 0.0360 |
| | | $13.5_3$ | 0.0000 | 0.9591 (2.6141) | 0.0409 | 0.0000 | 0.9729 (3.5759) | 0.0271 | 0.0000 | 0.9578 (3.6783) | 0.0422 | 0.0000 | 0.9819 (3.4867) | 0.0181 | 0.0000 | 0.9706 (3.5952) | 0.0294 |
| | $0.5_3$ | $5.0_3$ | 0.0001 | 0.9501 (1.1504) | 0.0498 | 0.0000 | 0.9772 (1.5916) | 0.0228 | 0.0000 | 0.9643 (1.5923) | 0.0357 | 0.0000 | 0.9760 (1.5779) | 0.0240 | 0.0000 | 0.9643 (1.5786) | 0.0357 |
| | | $5.5_3$ | 0.0003 | 0.9617 (1.2718) | 0.0380 | 0.0000 | 0.9837 (1.7612) | 0.0163 | 0.0000 | 0.9760 (1.7620) | 0.0240 | 0.0001 | 0.9829 (1.7459) | 0.0170 | 0.0000 | 0.9747 (1.7468) | 0.0253 |
| | | $6.0_3$ | 0.0000 | 0.9691 (1.3934) | 0.0309 | 0.0000 | 0.9881 (1.9312) | 0.0119 | 0.0000 | 0.9822 (1.9319) | 0.0178 | 0.0000 | 0.9871 (1.9146) | 0.0129 | 0.0000 | 0.9813 (1.9153) | 0.0187 |
| | | $6.5_3$ | 0.0001 | 0.9781 (1.5191) | 0.0218 | 0.0000 | 0.9937 (2.1050) | 0.0063 | 0.0000 | 0.9897 (2.1058) | 0.0103 | 0.0001 | 0.9930 (2.0869) | 0.0069 | 0.0000 | 0.9881 (2.0875) | 0.0119 |
| | $0.7_3$ | $3.0_3$ | 0.0009 | 0.9209 (0.6162) | 0.0782 | 0.0004 | 0.9680 (0.8653) | 0.0316 | 0.0002 | 0.9564 (0.8541) | 0.0434 | 0.0004 | 0.9618 (0.8542) | 0.0378 | 0.0000 | 0.9449 (0.8424) | 0.0551 |
| | | $3.5_3$ | 0.0010 | 0.9432 (0.7282) | 0.0558 | 0.0000 | 0.9792 (1.0243) | 0.0208 | 0.0000 | 0.9704 (1.0109) | 0.0296 | 0.0001 | 0.9743 (1.0112) | 0.0256 | 0.0000 | 0.9632 (0.9972) | 0.0368 |
| | | $4.0_3$ | 0.0003 | 0.9579 (0.8382) | 0.0418 | 0.0001 | 0.9889 (1.1802) | 0.0110 | 0.0000 | 0.9818 (1.1650) | 0.0182 | 0.0000 | 0.9858 (1.1652) | 0.0142 | 0.0000 | 0.9755 (1.1492) | 0.0245 |
| | | $4.5_3$ | 0.0002 | 0.9723 (0.9536) | 0.0275 | 0.0000 | 0.9929 (1.3420) | 0.0071 | 0.0000 | 0.9878 (1.3244) | 0.0122 | 0.0000 | 0.9895 (1.3249) | 0.0105 | 0.0000 | 0.9835 (1.3063) | 0.0165 |
| $50_3$ | $0.2_3$ | $12.0_3$ | 0.0000 | 0.9496 (1.8086) | 0.0504 | 0.0000 | 0.9622 (2.5059) | 0.0378 | 0.0000 | 0.9460 (2.5511) | 0.0540 | 0.0000 | 0.9717 (2.4595) | 0.0283 | 0.0000 | 0.9587 (2.5063) | 0.0413 |
| | | $12.5_3$ | 0.0000 | 0.9503 (1.8866) | 0.0497 | 0.0000 | 0.9685 (2.6136) | 0.0315 | 0.0000 | 0.9549 (2.6605) | 0.0451 | 0.0002 | 0.9772 (2.5649) | 0.0226 | 0.0000 | 0.9661 (2.6139) | 0.0339 |
| | | $13.0_3$ | 0.0000 | 0.9557 (1.9636) | 0.0443 | 0.0000 | 0.9718 (2.7213) | 0.0282 | 0.0000 | 0.9567 (2.7698) | 0.0433 | 0.0001 | 0.9796 (2.6705) | 0.0203 | 0.0000 | 0.9682 (2.7213) | 0.0318 |
| | | $13.5_3$ | 0.0000 | 0.9663 (2.0414) | 0.0337 | 0.0000 | 0.9769 (2.8280) | 0.0231 | 0.0000 | 0.9650 (2.8785) | 0.0350 | 0.0000 | 0.9833 (2.7752) | 0.0167 | 0.0000 | 0.9744 (2.8281) | 0.0256 |

**Table 1.** *Cont.*

| $n_i$ | $\delta_i$ | $a_i$ | FGCI | | | Baye.J | | | Baye.U | | | HPD.J | | | HPD.U | | |
|---|---|---|---|---|---|---|---|---|---|---|---|---|---|---|---|---|---|
| | | | LEP | CP (EL) | UEP | LEP | CP (EL) | UEP | LEP | CP (EL) | UEP | LEP | CP (EL) | UEP | LEP | CP (EL) | UEP |
| $50_3$ | $0.5_3$ | $5.0_3$ | 0.0000 | 0.9489 (0.9017) | 0.0511 | 0.0000 | 0.9764 (1.2595) | 0.0236 | 0.0000 | 0.9679 (1.2592) | 0.0321 | 0.0000 | 0.9755 (1.2483) | 0.0245 | 0.0000 | 0.9650 (1.2483) | 0.0350 |
| | | $5.5_3$ | 0.0000 | 0.9610 (0.9968) | 0.0390 | 0.0000 | 0.9835 (1.3904) | 0.0165 | 0.0000 | 0.9759 (1.3903) | 0.0241 | 0.0000 | 0.9829 (1.3783) | 0.0171 | 0.0000 | 0.9745 (1.3782) | 0.0255 |
| | | $6.0_3$ | 0.0000 | 0.9744 (1.0920) | 0.0256 | 0.0000 | 0.9897 (1.5256) | 0.0103 | 0.0000 | 0.9844 (1.5253) | 0.0156 | 0.0000 | 0.9883 (1.5121) | 0.0117 | 0.0000 | 0.9832 (1.5121) | 0.0168 |
| | | $6.5_3$ | 0.0000 | 0.9810 (1.1899) | 0.0190 | 0.0001 | 0.9926 (1.6589) | 0.0073 | 0.0000 | 0.9878 (1.6587) | 0.0122 | 0.0001 | 0.9920 (1.6442) | 0.0079 | 0.0000 | 0.9862 (1.6441) | 0.0138 |
| | $0.7_3$ | $3.0_3$ | 0.0004 | 0.9119 (0.4784) | 0.0877 | 0.0003 | 0.9667 (0.6746) | 0.0330 | 0.0001 | 0.9526 (0.6687) | 0.0473 | 0.0002 | 0.9587 (0.6668) | 0.0411 | 0.0001 | 0.9446 (0.6608) | 0.0553 |
| | | $3.5_3$ | 0.0005 | 0.9423 (0.5665) | 0.0572 | 0.0000 | 0.9792 (0.7984) | 0.0208 | 0.0000 | 0.9697 (0.7916) | 0.0303 | 0.0000 | 0.9736 (0.7891) | 0.0264 | 0.0000 | 0.9624 (0.7821) | 0.0376 |
| | | $4.0_3$ | 0.0001 | 0.9679 (0.6546) | 0.0320 | 0.0000 | 0.9881 (0.9229) | 0.0119 | 0.0000 | 0.9831 (0.9148) | 0.0169 | 0.0000 | 0.9851 (0.9120) | 0.0149 | 0.0000 | 0.9786 (0.9039) | 0.0214 |
| | | $4.5_3$ | 0.0001 | 0.9770 (0.7427) | 0.0229 | 0.0000 | 0.9929 (1.0481) | 0.0071 | 0.0000 | 0.9892 (1.0391) | 0.0108 | 0.0000 | 0.9900 (1.0359) | 0.0100 | 0.0000 | 0.9855 (1.0266) | 0.0145 |
| $100_3$ | $0.2_3$ | $12.0_3$ | 0.0000 | 0.9343 (1.2868) | 0.0657 | 0.0000 | 0.9556 (1.8010) | 0.0444 | 0.0000 | 0.9408 (1.8172) | 0.0592 | 0.0000 | 0.9639 (1.7760) | 0.0361 | 0.0000 | 0.9513 (1.7926) | 0.0487 |
| | | $12.5_3$ | 0.0000 | 0.9465 (1.3415) | 0.0535 | 0.0000 | 0.9629 (1.8780) | 0.0371 | 0.0000 | 0.9504 (1.8950) | 0.0496 | 0.0000 | 0.9693 (1.8521) | 0.0307 | 0.0000 | 0.9593 (1.8691) | 0.0407 |
| | | $13.0_3$ | 0.0000 | 0.9569 (1.3969) | 0.0431 | 0.0000 | 0.9673 (1.9550) | 0.0327 | 0.0000 | 0.9548 (1.9729) | 0.0452 | 0.0000 | 0.9736 (1.9280) | 0.0264 | 0.0000 | 0.9633 (1.9460) | 0.0367 |
| | | $13.5_3$ | 0.0000 | 0.9647 (1.4520) | 0.0353 | 0.0000 | 0.9718 (2.0316) | 0.0282 | 0.0000 | 0.9593 (2.0500) | 0.0407 | 0.0001 | 0.9779 (2.0035) | 0.0220 | 0.0000 | 0.9662 (2.0223) | 0.0338 |
| | $0.5_3$ | $5.0_3$ | 0.0000 | 0.9221 (0.6429) | 0.0779 | 0.0000 | 0.9571 (0.9018) | 0.0429 | 0.0000 | 0.9462 (0.9019) | 0.0538 | 0.0000 | 0.9540 (0.8938) | 0.0460 | 0.0000 | 0.9413 (0.8939) | 0.0587 |
| | | $5.5_3$ | 0.0000 | 0.9444 (0.7110) | 0.0556 | 0.0000 | 0.9758 (0.9983) | 0.0242 | 0.0000 | 0.9679 (0.9984) | 0.0321 | 0.0000 | 0.9744 (0.9894) | 0.0256 | 0.0000 | 0.9658 (0.9895) | 0.0342 |
| | | $6.0_3$ | 0.0000 | 0.9667 (0.7789) | 0.0333 | 0.0000 | 0.9824 (1.0946) | 0.0176 | 0.0000 | 0.9767 (1.0948) | 0.0233 | 0.0000 | 0.9820 (1.0849) | 0.0180 | 0.0000 | 0.9742 (1.0850) | 0.0258 |
| | | $6.5_3$ | 0.0000 | 0.9787 (0.8484) | 0.0213 | 0.0000 | 0.9893 (1.1929) | 0.0107 | 0.0000 | 0.9858 (1.1931) | 0.0142 | 0.0000 | 0.9895 (1.1822) | 0.0105 | 0.0000 | 0.9850 (1.1825) | 0.0150 |

**Table 1.** *Cont.*

| $n_i$ | $\delta_i$ | $a_i$ | FGCI LEP | FGCI CP (EL) | FGCI UEP | Baye.J LEP | Baye.J CP (EL) | Baye.J UEP | Baye.U LEP | Baye.U CP (EL) | Baye.U UEP | HPD.J LEP | HPD.J CP (EL) | HPD.J UEP | HPD.U LEP | HPD.U CP (EL) | HPD.U UEP |
|---|---|---|---|---|---|---|---|---|---|---|---|---|---|---|---|---|---|
| $100_3$ | $0.7_3$ | $3.0_3$ | 0.0001 | 0.8809 (0.3393) | 0.1190 | 0.0001 | 0.9449 (0.4797) | 0.0550 | 0.0000 | 0.9319 (0.4776) | 0.0681 | 0.0000 | 0.9358 (0.4747) | 0.0642 | 0.0000 | 0.9205 (0.4726) | 0.0795 |
| | | $3.5_3$ | 0.0000 | 0.9280 (0.4022) | 0.0720 | 0.0000 | 0.9691 (0.5677) | 0.0309 | 0.0000 | 0.9594 (0.5652) | 0.0406 | 0.0001 | 0.9622 (0.5618) | 0.0377 | 0.0000 | 0.9505 (0.5592) | 0.0495 |
| | | $4.0_3$ | 0.0000 | 0.9579 (0.4650) | 0.0421 | 0.0000 | 0.9819 (0.6564) | 0.0181 | 0.0000 | 0.9762 (0.6534) | 0.0238 | 0.0000 | 0.9780 (0.6495) | 0.0220 | 0.0000 | 0.9695 (0.6466) | 0.0305 |
| | | $4.5_3$ | 0.0000 | 0.9737 (0.5282) | 0.0263 | 0.0000 | 0.9911 (0.7461) | 0.0089 | 0.0000 | 0.9867 (0.7429) | 0.0133 | 0.0000 | 0.9889 (0.7384) | 0.0111 | 0.0000 | 0.9824 (0.7351) | 0.0176 |
| $30_1$, $50_1$, $100_1$ | $0.2_3$ | $12.0_3$ | 0.0000 | 0.9882 (2.3137) | 0.0118 | 0.0000 | 0.9581 (3.1706) | 0.0419 | 0.0000 | 0.9408 (3.2610) | 0.0592 | 0.0000 | 0.9706 (3.0919) | 0.0294 | 0.0000 | 0.9554 (3.1877) | 0.0446 |
| | | $12.5_3$ | 0.0000 | 0.9906 (2.4129) | 0.0094 | 0.0000 | 0.9650 (3.3027) | 0.0350 | 0.0000 | 0.9450 (3.3973) | 0.0550 | 0.0000 | 0.9751 (3.2204) | 0.0249 | 0.0000 | 0.9587 (3.3207) | 0.0413 |
| | | $13.0_3$ | 0.0000 | 0.9933 (2.5129) | 0.0067 | 0.0000 | 0.9710 (3.4409) | 0.0290 | 0.0000 | 0.9530 (3.5386) | 0.0470 | 0.0002 | 0.9804 (3.3551) | 0.0194 | 0.0000 | 0.9667 (3.4588) | 0.0333 |
| | | $13.5_3$ | 0.0000 | 0.9931 (2.6116) | 0.0069 | 0.0000 | 0.9736 (3.5757) | 0.0264 | 0.0000 | 0.9575 (3.6782) | 0.0425 | 0.0000 | 0.9808 (3.4871) | 0.0192 | 0.0000 | 0.9699 (3.5955) | 0.0301 |
| | $0.5_3$ | $5.0_3$ | 0.0000 | 0.9933 (1.1438) | 0.0067 | 0.0000 | 0.9766 (1.5928) | 0.0234 | 0.0000 | 0.9662 (1.5933) | 0.0338 | 0.0001 | 0.9756 (1.5791) | 0.0243 | 0.0000 | 0.9643 (1.5796) | 0.0357 |
| | | $5.5_3$ | 0.0000 | 0.9960 (1.2663) | 0.0040 | 0.0000 | 0.9856 (1.7614) | 0.0144 | 0.0000 | 0.9784 (1.7622) | 0.0216 | 0.0000 | 0.9848 (1.7463) | 0.0152 | 0.0000 | 0.9772 (1.7470) | 0.0228 |
| | | $6.0_3$ | 0.0000 | 0.9983 (1.3889) | 0.0017 | 0.0000 | 0.9901 (1.9300) | 0.0099 | 0.0000 | 0.9845 (1.9309) | 0.0155 | 0.0000 | 0.9890 (1.9134) | 0.0110 | 0.0000 | 0.9835 (1.9142) | 0.0165 |
| | | $6.5_3$ | 0.0000 | 0.9987 (1.5109) | 0.0013 | 0.0000 | 0.9916 (2.0987) | 0.0084 | 0.0000 | 0.9860 (2.0995) | 0.0140 | 0.0000 | 0.9909 (2.0807) | 0.0091 | 0.0000 | 0.9851 (2.0814) | 0.0149 |
| | $0.7_3$ | $3.0_3$ | 0.0000 | 0.9867 (0.6079) | 0.0113 | 0.0003 | 0.9676 (0.8625) | 0.0321 | 0.0000 | 0.9545 (0.8511) | 0.0455 | 0.0002 | 0.9608 (0.8514) | 0.0390 | 0.0000 | 0.9444 (0.8394) | 0.0556 |
| | | $3.5_3$ | 0.0000 | 0.9927 (0.7194) | 0.0073 | 0.0000 | 0.9818 (1.0253) | 0.0182 | 0.0000 | 0.9743 (1.0120) | 0.0257 | 0.0000 | 0.9765 (1.0123) | 0.0235 | 0.0000 | 0.9659 (0.9982) | 0.0341 |
| | | $4.0_3$ | 0.0000 | 0.9973 (0.8318) | 0.0027 | 0.0002 | 0.9880 (1.1796) | 0.0118 | 0.0000 | 0.9811 (1.1642) | 0.0189 | 0.0001 | 0.9843 (1.1646) | 0.0156 | 0.0000 | 0.9752 (1.1485) | 0.0248 |
| | | $4.5_3$ | 0.0000 | 0.9986 (0.9441) | 0.0014 | 0.0000 | 0.9930 (1.3400) | 0.0070 | 0.0000 | 0.9889 (1.3225) | 0.0111 | 0.0000 | 0.9917 (1.3230) | 0.0083 | 0.0000 | 0.9845 (1.3046) | 0.0155 |

**Table 2.** The coverage probabilities and expected lengths for estimating the 95% confidence interval for the common mean of several ZIG distributions ($k = 5$).

| $n_i$ | $\delta_i$ | $a_i$ | FGCI | | | Baye.J | | | Baye.U | | | HPD.J | | | HPD.U | | |
|---|---|---|---|---|---|---|---|---|---|---|---|---|---|---|---|---|---|
| | | | LEP | CP (EL) | UEP | LEP | CP (EL) | UEP | LEP | CP (EL) | UEP | LEP | CP (EL) | UEP | LEP | CP (EL) | UEP |
| $30_5$ | $0.2_5$ | $12.0_5$ | 0.0000 | 0.9628 (2.3170) | 0.0372 | 0.0000 | 0.9635 (3.1723) | 0.0365 | 0.0000 | 0.9426 (3.2623) | 0.0574 | 0.0000 | 0.9738 (3.0934) | 0.0262 | 0.0000 | 0.9601 (3.1891) | 0.0399 |
| | | $12.5_5$ | 0.0000 | 0.9720 (2.4154) | 0.0280 | 0.0000 | 0.9637 (3.3088) | 0.0363 | 0.0000 | 0.9448 (3.4036) | 0.0552 | 0.0000 | 0.9753 (3.2267) | 0.0247 | 0.0000 | 0.9597 (3.3268) | 0.0403 |
| | | $13.0_5$ | 0.0000 | 0.9742 (2.5154) | 0.0258 | 0.0000 | 0.9666 (3.4382) | 0.0334 | 0.0000 | 0.9494 (3.5362) | 0.0506 | 0.0000 | 0.9763 (3.3527) | 0.0237 | 0.0000 | 0.9627 (3.4567) | 0.0373 |
| | | $13.5_5$ | 0.0000 | 0.9789 (2.6120) | 0.0211 | 0.0000 | 0.9735 (3.5766) | 0.0265 | 0.0000 | 0.9569 (3.6786) | 0.0431 | 0.0000 | 0.9810 (3.4876) | 0.0190 | 0.0000 | 0.9708 (3.5955) | 0.0292 |
| | $0.5_5$ | $5.0_5$ | 0.0000 | 0.9679 (1.1513) | 0.0321 | 0.0001 | 0.9767 (1.5929) | 0.0232 | 0.0000 | 0.9668 (1.5938) | 0.0332 | 0.0001 | 0.9762 (1.5792) | 0.0237 | 0.0000 | 0.9645 (1.5801) | 0.0355 |
| | | $5.5_5$ | 0.0000 | 0.9766 (1.2706) | 0.0234 | 0.0000 | 0.9856 (1.7634) | 0.0144 | 0.0000 | 0.9779 (1.7641) | 0.0221 | 0.0000 | 0.9846 (1.7482) | 0.0154 | 0.0000 | 0.9768 (1.7489) | 0.0232 |
| | | $6.0_5$ | 0.0000 | 0.9798 (1.3934) | 0.0202 | 0.0000 | 0.9886 (1.9328) | 0.0114 | 0.0000 | 0.9818 (1.9334) | 0.0182 | 0.0000 | 0.9888 (1.9160) | 0.0112 | 0.0000 | 0.9811 (1.9168) | 0.0189 |
| | | $6.5_5$ | 0.0000 | 0.9857 (1.5139) | 0.0143 | 0.0000 | 0.9926 (2.1022) | 0.0074 | 0.0000 | 0.9875 (2.1029) | 0.0125 | 0.0000 | 0.9917 (2.0838) | 0.0083 | 0.0000 | 0.9869 (2.0849) | 0.0131 |
| | $0.7_5$ | $3.0_5$ | 0.0005 | 0.9364 (0.6131) | 0.0631 | 0.0004 | 0.9693 (0.8649) | 0.0303 | 0.0001 | 0.9555 (0.8538) | 0.0444 | 0.0003 | 0.9623 (0.8539) | 0.0374 | 0.0000 | 0.9445 (0.8421) | 0.0555 |
| | | $3.5_5$ | 0.0002 | 0.9590 (0.7270) | 0.0408 | 0.0001 | 0.9805 (1.0219) | 0.0194 | 0.0000 | 0.9723 (1.0087) | 0.0277 | 0.0000 | 0.9763 (1.0089) | 0.0237 | 0.0000 | 0.9662 (0.9949) | 0.0338 |
| | | $4.0_5$ | 0.0004 | 0.9718 (0.8387) | 0.0278 | 0.0000 | 0.9887 (1.1836) | 0.0113 | 0.0000 | 0.9824 (1.1681) | 0.0176 | 0.0000 | 0.9855 (1.1687) | 0.0145 | 0.0000 | 0.9773 (1.1522) | 0.0227 |
| | | $4.5_5$ | 0.0000 | 0.9813 (0.9526) | 0.0187 | 0.0000 | 0.9939 (1.3418) | 0.0061 | 0.0000 | 0.9891 (1.3247) | 0.0109 | 0.0000 | 0.9909 (1.3247) | 0.0091 | 0.0000 | 0.9853 (1.3067) | 0.0147 |
| $50_5$ | $0.2_5$ | $12.0_5$ | 0.0000 | 0.9731 (1.8091) | 0.0269 | 0.0001 | 0.9635 (2.5072) | 0.0364 | 0.0000 | 0.9486 (2.5521) | 0.0514 | 0.0003 | 0.9742 (2.4605) | 0.0255 | 0.0001 | 0.9603 (2.5074) | 0.0396 |
| | | $12.5_5$ | 0.0000 | 0.9789 (1.8858) | 0.0211 | 0.0001 | 0.9635 (2.6123) | 0.0364 | 0.0001 | 0.9501 (2.6591) | 0.0498 | 0.0001 | 0.9723 (2.5636) | 0.0276 | 0.0001 | 0.9587 (2.6125) | 0.0412 |
| | | $13.0_5$ | 0.0001 | 0.9808 (1.9636) | 0.0191 | 0.0001 | 0.9703 (2.7208) | 0.0296 | 0.0000 | 0.9562 (2.7696) | 0.0438 | 0.0002 | 0.9789 (2.6704) | 0.0209 | 0.0000 | 0.9659 (2.7212) | 0.0341 |
| | | $13.5_5$ | 0.0000 | 0.9843 (2.0403) | 0.0157 | 0.0000 | 0.9773 (2.8274) | 0.0227 | 0.0000 | 0.9647 (2.8779) | 0.0353 | 0.0000 | 0.9831 (2.7747) | 0.0169 | 0.0000 | 0.9746 (2.8275) | 0.0254 |

**Table 2.** *Cont.*

| $n_i$ | $\delta_i$ | $a_i$ | FGCI | | | Baye.J | | | Baye.U | | | HPD.J | | | HPD.U | | |
|---|---|---|---|---|---|---|---|---|---|---|---|---|---|---|---|---|---|
| | | | LEP | CP (EL) | UEP | LEP | CP (EL) | UEP | LEP | CP (EL) | UEP | LEP | CP (EL) | UEP | LEP | CP (EL) | UEP |
| $50_5$ | | $5.0_5$ | 0.0000 | 0.9710 (0.9010) | 0.0290 | 0.0001 | 0.9752 (1.2582) | 0.0247 | 0.0000 | 0.9664 (1.2578) | 0.0336 | 0.0000 | 0.9746 (1.2470) | 0.0254 | 0.0000 | 0.9647 (1.2469) | 0.0353 |
| | | $5.5_5$ | 0.0000 | 0.9809 (0.9975) | 0.0191 | 0.0000 | 0.9835 (1.3924) | 0.0165 | 0.0000 | 0.9762 (1.3922) | 0.0238 | 0.0000 | 0.9829 (1.3801) | 0.0171 | 0.0000 | 0.9735 (1.3800) | 0.0265 |
| | $0.5_5$ | $6.0_5$ | 0.0000 | 0.9883 (1.0926) | 0.0117 | 0.0000 | 0.9902 (1.5265) | 0.0098 | 0.0000 | 0.9850 (1.5264) | 0.0150 | 0.0000 | 0.9888 (1.5132) | 0.0112 | 0.0000 | 0.9833 (1.5131) | 0.0167 |
| | | $6.5_5$ | 0.0000 | 0.9916 (1.1882) | 0.0084 | 0.0000 | 0.9910 (1.6601) | 0.0090 | 0.0000 | 0.9870 (1.6599) | 0.0130 | 0.0000 | 0.9910 (1.6455) | 0.0090 | 0.0000 | 0.9841 (1.6454) | 0.0159 |
| | | $3.0_5$ | 0.0002 | 0.9414 (0.4788) | 0.0584 | 0.0001 | 0.9607 (0.6763) | 0.0392 | 0.0000 | 0.9493 (0.6704) | 0.0507 | 0.0001 | 0.9536 (0.6684) | 0.0463 | 0.0001 | 0.9383 (0.6623) | 0.0616 |
| | | $3.5_5$ | 0.0004 | 0.9623 (0.5652) | 0.0373 | 0.0001 | 0.9806 (0.7997) | 0.0193 | 0.0001 | 0.9717 (0.7928) | 0.0282 | 0.0001 | 0.9753 (0.7903) | 0.0246 | 0.0000 | 0.9651 (0.7833) | 0.0349 |
| | $0.7_5$ | $4.0_5$ | 0.0001 | 0.9790 (0.6544) | 0.0209 | 0.0001 | 0.9862 (0.9207) | 0.0137 | 0.0000 | 0.9817 (0.9128) | 0.0183 | 0.0001 | 0.9834 (0.9099) | 0.0165 | 0.0000 | 0.9759 (0.9019) | 0.0241 |
| | | $4.5_5$ | 0.0000 | 0.9865 (0.7422) | 0.0135 | 0.0000 | 0.9938 (1.0485) | 0.0062 | 0.0000 | 0.9892 (1.0394) | 0.0108 | 0.0000 | 0.9907 (1.0364) | 0.0093 | 0.0000 | 0.9861 (1.0270) | 0.0139 |
| $100_5$ | | $12.0_5$ | 0.0000 | 0.9719 (1.2869) | 0.0281 | 0.0000 | 0.9560 (1.8007) | 0.0440 | 0.0000 | 0.9426 (1.8171) | 0.0574 | 0.0000 | 0.9637 (1.7760) | 0.0363 | 0.0000 | 0.9514 (1.7926) | 0.0486 |
| | | $12.5_5$ | 0.0000 | 0.9734 (1.3415) | 0.0266 | 0.0001 | 0.9660 (1.8788) | 0.0339 | 0.0000 | 0.9534 (1.8959) | 0.0466 | 0.0001 | 0.9717 (1.8529) | 0.0282 | 0.0001 | 0.9615 (1.8703) | 0.0384 |
| | $0.2_5$ | $13.0_5$ | 0.0000 | 0.9835 (1.3968) | 0.0165 | 0.0000 | 0.9695 (1.9536) | 0.0305 | 0.0000 | 0.9547 (1.9713) | 0.0453 | 0.0000 | 0.9754 (1.9267) | 0.0264 | 0.0000 | 0.9624 (1.9444) | 0.0376 |
| | | $13.5_5$ | 0.0000 | 0.9852 (1.4518) | 0.0148 | 0.0000 | 0.9734 (2.0305) | 0.0266 | 0.0000 | 0.9603 (2.0489) | 0.0397 | 0.0000 | 0.9782 (2.0024) | 0.0218 | 0.0000 | 0.9670 (2.0212) | 0.0330 |
| | | $5.0_5$ | 0.0000 | 0.9540 (0.6422) | 0.0460 | 0.0000 | 0.9610 (0.9019) | 0.0390 | 0.0000 | 0.9501 (0.9022) | 0.0499 | 0.0000 | 0.9584 (0.8940) | 0.0416 | 0.0000 | 0.9473 (0.8942) | 0.0527 |
| | | $5.5_5$ | 0.0000 | 0.9763 (0.7111) | 0.0237 | 0.0000 | 0.9761 (0.9976) | 0.0239 | 0.0000 | 0.9680 (0.9978) | 0.0320 | 0.0000 | 0.9735 (0.9888) | 0.0265 | 0.0000 | 0.9649 (0.9890) | 0.0351 |
| | $0.5_5$ | $6.0_5$ | 0.0000 | 0.9860 (0.7796) | 0.0140 | 0.0000 | 0.9825 (1.0957) | 0.0175 | 0.0000 | 0.9764 (1.0959) | 0.0236 | 0.0000 | 0.9822 (1.0860) | 0.0178 | 0.0000 | 0.9745 (1.0861) | 0.0255 |
| | | $6.5_5$ | 0.0000 | 0.9893 (0.8478) | 0.0107 | 0.0000 | 0.9902 (1.1909) | 0.0098 | 0.0000 | 0.9855 (1.1910) | 0.0145 | 0.0000 | 0.9886 (1.1803) | 0.0114 | 0.0000 | 0.9846 (1.1805) | 0.0154 |

**Table 2.** *Cont.*

| $n_i$ | $\delta_i$ | $a_i$ | FGCI | | | Baye.J | | | Baye.U | | | HPD.J | | | HPD.U | | |
|---|---|---|---|---|---|---|---|---|---|---|---|---|---|---|---|---|---|
| | | | LEP | CP (EL) | UEP | LEP | CP (EL) | UEP | LEP | CP (EL) | UEP | LEP | CP (EL) | UEP | LEP | CP (EL) | UEP |
| $100_5$ | $0.7_5$ | $3.0_5$ | 0.0000 | 0.9151 (0.3393) | 0.0849 | 0.0000 | 0.9445 (0.4795) | 0.0555 | 0.0000 | 0.9308 (0.4773) | 0.0692 | 0.0000 | 0.9350 (0.4744) | 0.0650 | 0.0000 | 0.9206 (0.4723) | 0.0794 |
| | | $3.5_5$ | 0.0000 | 0.9563 (0.4017) | 0.0437 | 0.0000 | 0.9707 (0.5678) | 0.0293 | 0.0000 | 0.9609 (0.5654) | 0.0391 | 0.0000 | 0.9659 (0.5619) | 0.0341 | 0.0000 | 0.9533 (0.5594) | 0.0467 |
| | | $4.0_5$ | 0.0001 | 0.9777 (0.4651) | 0.0222 | 0.0000 | 0.9839 (0.6570) | 0.0161 | 0.0000 | 0.9784 (0.6542) | 0.0216 | 0.0000 | 0.9794 (0.6501) | 0.0206 | 0.0000 | 0.9725 (0.6473) | 0.0275 |
| | | $4.5_5$ | 0.0000 | 0.9885 (0.5279) | 0.0115 | 0.0000 | 0.9908 (0.7459) | 0.0092 | 0.0000 | 0.9865 (0.7426) | 0.0135 | 0.0000 | 0.9883 (0.7381) | 0.0117 | 0.0000 | 0.9838 (0.7347) | 0.0162 |
| $30_2,$ $50_1,$ $100_2$ | $0.2_5$ | $12.0_5$ | 0.0000 | 0.9975 (2.3126) | 0.0025 | 0.0000 | 0.9614 (3.1679) | 0.0386 | 0.0000 | 0.9395 (3.2581) | 0.0605 | 0.0000 | 0.9734 (3.0894) | 0.0266 | 0.0000 | 0.9556 (3.1849) | 0.0444 |
| | | $12.5_5$ | 0.0000 | 0.9980 (2.4128) | 0.0020 | 0.0000 | 0.9683 (3.3074) | 0.0317 | 0.0000 | 0.9500 (3.4016) | 0.0500 | 0.0000 | 0.9774 (3.2251) | 0.0226 | 0.0000 | 0.9629 (3.3251) | 0.0371 |
| | | $13.0_5$ | 0.0000 | 0.9985 (2.5123) | 0.0015 | 0.0000 | 0.9695 (3.4399) | 0.0305 | 0.0000 | 0.9511 (3.5379) | 0.0489 | 0.0002 | 0.9797 (3.3544) | 0.0201 | 0.0000 | 0.9640 (3.4584) | 0.0360 |
| | | $13.5_5$ | 0.0000 | 0.9992 (2.6110) | 0.0008 | 0.0000 | 0.9741 (3.5761) | 0.0259 | 0.0000 | 0.9588 (3.6776) | 0.0412 | 0.0000 | 0.9814 (3.4869) | 0.0186 | 0.0000 | 0.9706 (3.5952) | 0.0294 |
| | $0.5_5$ | $5.0_5$ | 0.0000 | 0.9986 (1.1441) | 0.0014 | 0.0000 | 0.9773 (1.5931) | 0.0227 | 0.0000 | 0.9670 (1.5937) | 0.0330 | 0.0001 | 0.9764 (1.5793) | 0.0235 | 0.0000 | 0.9663 (1.5801) | 0.0337 |
| | | $5.5_5$ | 0.0000 | 0.9985 (1.2666) | 0.0015 | 0.0000 | 0.9850 (1.7628) | 0.0150 | 0.0000 | 0.9767 (1.7635) | 0.0233 | 0.0001 | 0.9842 (1.7477) | 0.0157 | 0.0000 | 0.9758 (1.7483) | 0.0242 |
| | | $6.0_5$ | 0.0000 | 0.9995 (1.3882) | 0.0005 | 0.0000 | 0.9890 (1.9295) | 0.0110 | 0.0000 | 0.9811 (1.9305) | 0.0189 | 0.0000 | 0.9894 (1.9128) | 0.0106 | 0.0000 | 0.9792 (1.9138) | 0.0208 |
| | | $6.5_5$ | 0.0000 | 0.9997 (1.5113) | 0.0003 | 0.0000 | 0.9933 (2.1024) | 0.0067 | 0.0000 | 0.9898 (2.1034) | 0.0102 | 0.0000 | 0.9931 (2.0844) | 0.0069 | 0.0000 | 0.9887 (2.0853) | 0.0113 |
| | $0.7_5$ | $3.0_5$ | 0.0001 | 0.9938 (0.6063) | 0.0061 | 0.0003 | 0.9678 (0.8630) | 0.0319 | 0.0000 | 0.9541 (0.8518) | 0.0459 | 0.0002 | 0.9609 (0.8520) | 0.0389 | 0.0000 | 0.9449 (0.8403) | 0.0551 |
| | | $3.5_5$ | 0.0000 | 0.9968 (0.7187) | 0.0032 | 0.0000 | 0.9799 (1.0241) | 0.0201 | 0.0000 | 0.9686 (1.0109) | 0.0314 | 0.0000 | 0.9742 (1.0110) | 0.0258 | 0.0000 | 0.9614 (0.9970) | 0.0386 |
| | | $4.0_5$ | 0.0000 | 0.9995 (0.8306) | 0.0005 | 0.0003 | 0.9880 (1.1842) | 0.0117 | 0.0000 | 0.9831 (1.1685) | 0.0169 | 0.0000 | 0.9855 (1.1691) | 0.0145 | 0.0000 | 0.9762 (1.1526) | 0.0238 |
| | | $4.5_5$ | 0.0000 | 0.9997 (0.9438) | 0.0003 | 0.0000 | 0.9930 (1.3417) | 0.0070 | 0.0000 | 0.9883 (1.3242) | 0.0117 | 0.0000 | 0.9907 (1.3246) | 0.0093 | 0.0000 | 0.9851 (1.3062) | 0.0149 |

We also plotted the coverage probabilities and expected lengths for the five confidence interval estimation methods for scenarios with various sample sizes and probabilities of zero values in Figures 1–4. For $k = 3$, in almost all cases, the coverage probabilities of all of the proposed methods were close to or greater than the nominal confidence level of 0.95, while the expected length of the fiducial GCI was the shortest. However, in some cases, the fiducial GCI method was marginally outperformed by the HPD interval based on the Jeffreys rule or uniform prior. The results were similar for $k = 5$, although the fiducial GCI method obtained coverage probabilities greater than 0.95 in all cases, which was better than for $k = 3$. For unequal sample sizes, the fiducial GCI method obtained coverage probabilities greater than 0.95 even though their expected lengths were shorter than the others in all case for $k = 3$ and 5. For equal sample sizes, the coverage probabilities of the fiducial GCI were less than the nominal confidence level 0.95 in some case for $k = 3$ and 5. According to the results from Tables 1 and 2, the tail error rate of the proposed methods were unbalanced, whereas the expected length of the fiducial GCI was the smallest length of coverage probabilities over 0.95. When the fiducial GCI was less than 0.95, the HPD interval based on the Jeffreys rule or uniform prior outperformed the fiducial GCI. Therefore, the fiducial GCI and the HPD interval based on the Jeffreys rule or uniform prior should be used to compute the confidence interval estimation for the common mean of several ZIG distributions.

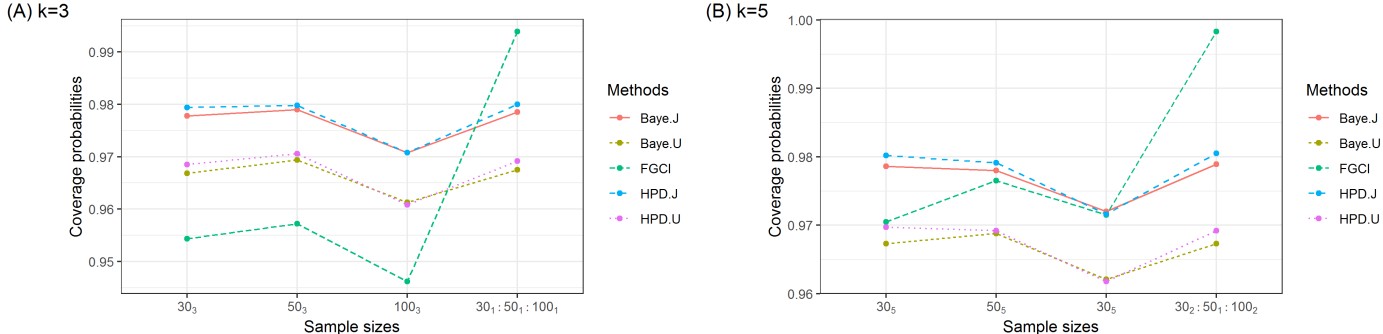

**Figure 1.** Comparison of the coverage probabilities for estimating the 95% confidence interval for the common mean of several ZIG distributions for various sample sizes: (**A**) $k = 3$ and (**B**) $k = 5$.

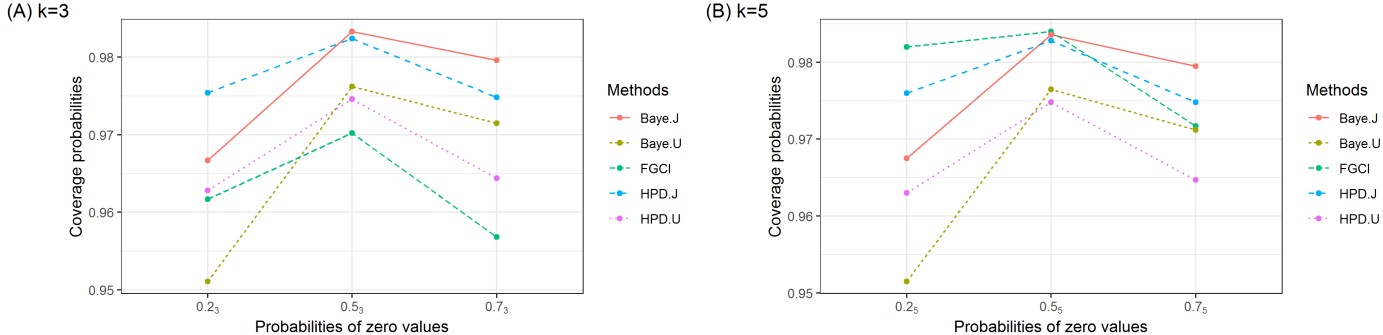

**Figure 2.** Comparison of the coverage probabilities for estimating the 95% confidence interval for the common mean of several ZIG distributions for various probabilities of zero values: (**A**) $k = 3$ and (**B**) $k = 5$.

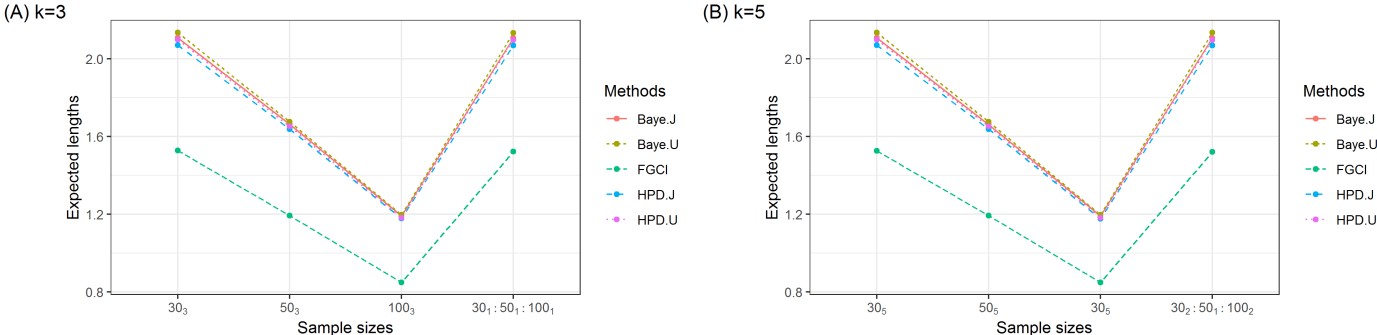

**Figure 3.** Comparison of expected lengths for estimation of the 95% confidence interval for the common mean of several ZIG distributions for various sample sizes: (**A**) $k = 3$ and (**B**) $k = 5$.

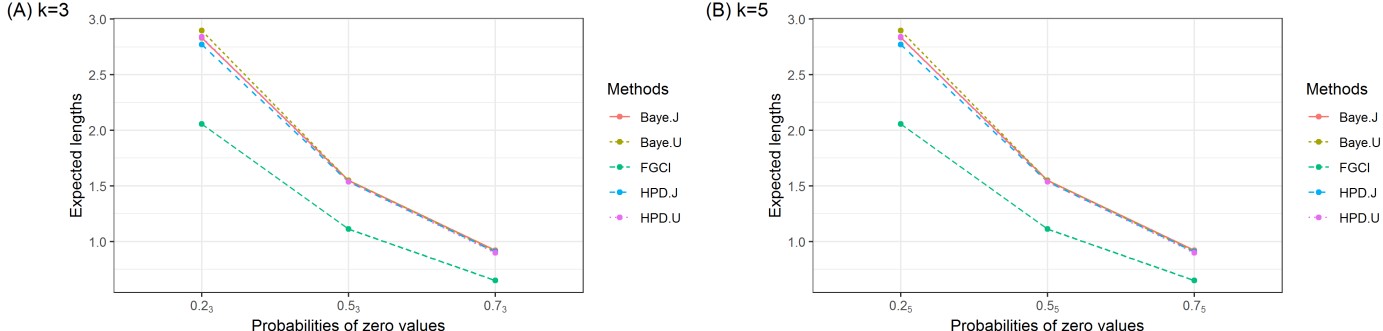

**Figure 4.** Comparison of the expected length for estimating the 95% confidence interval for the common mean of several ZIG distributions for various probabilities of zero values: (**A**) $k = 3$ and (**B**) $k = 5$.

### 3.2. Robustness Studies

In this section, we study the robustness of our proposed methods. In order to conduct the study, a small amount of random noise was added, and ZIG distributions were used to create the samples. We considered sample sizes of ($30_3$) and ($100_3$) with the settings given in Table 3.

**Table 3.** Different $\delta_i$ and $a_i$ values in robustness studies with $k = 3$.

| Setting | 1 | 2 | 3 | 4 | 5 | 6 | 7 | 8 | 9 | 10 | 11 | 12 |
|---|---|---|---|---|---|---|---|---|---|---|---|---|
| $\delta_i$ | $0.2_3$ | $0.2_3$ | $0.2_3$ | $0.2_3$ | $0.5_3$ | $0.5_3$ | $0.5_3$ | $0.5_3$ | $0.7_3$ | $0.7_3$ | $0.7_3$ | $0.7_3$ |
| $a_i$ | $12.0_3$ | $12.5_3$ | $13.0_3$ | $13.5_3$ | $5.0_3$ | $5.5_3$ | $6.0_3$ | $6.5_3$ | $3.0_3$ | $3.5_3$ | $4.0_3$ | $4.5_3$ |

From the results shown in Table 4, we notice that even with some noise added to the sample, the fiducial GCI, Bayesian, and HPD based on Jeffreys rule or uniform prior return satisfactory results according to coverage probabilities. Almost all coverage probabilities are slightly higher than the nominal level with the fiducial GCI, Bayesian, and HPD based on Jeffreys rule or uniform prior closer to or greater than the nominal level 0.95. The expected length values based on fiducial GCI are smaller than the Bayesian and HPD based on Jeffreys rule or uniform prior. The coverage probabilities and expected lengths of the confidence intervals based on the five methods are displayed in Figures 5 and 6. In addition, when the samples contain noise, the results from the fiducial GCI and HPD based on Jeffreys rule or uniform prior performed well. Therefore, the proposed methods seem robust to the samples that contain noise. In addition, robust statistics resist the influence of non-normal distributions; they perform well in a wide variety of probability distributions. In robustness study, data are generated from ZIG distributions, which are non-normal

distributions. When the sample size increases from $(30_3)$ to $(100_3)$, the proposed methods with respect to the departure from the ZIG distributions is robust, because the expected length of proposed methods became shorter.

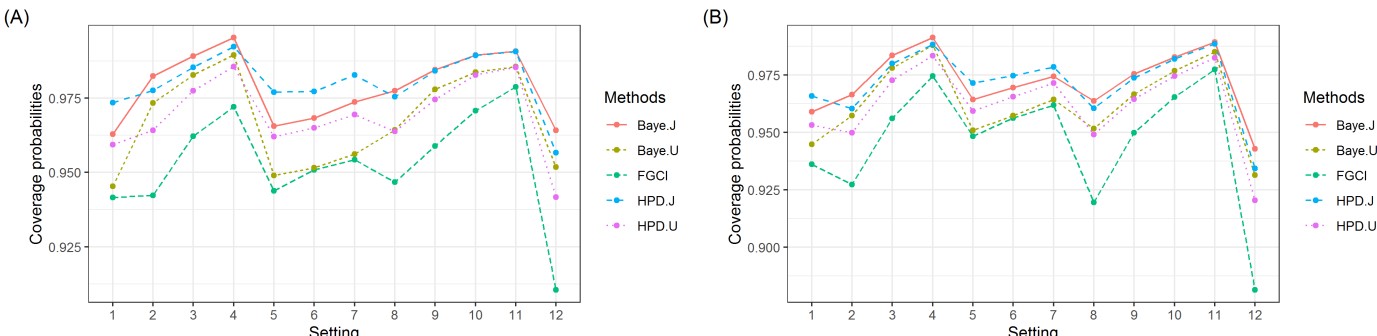

**Figure 5.** Comparison of the coverage probabilities for estimating the 95% confidence interval for the common mean of several ZIG distributions with the samples containing noise: (**A**) $n_i = 30_3$ and (**B**) $n_i = 100_3$.

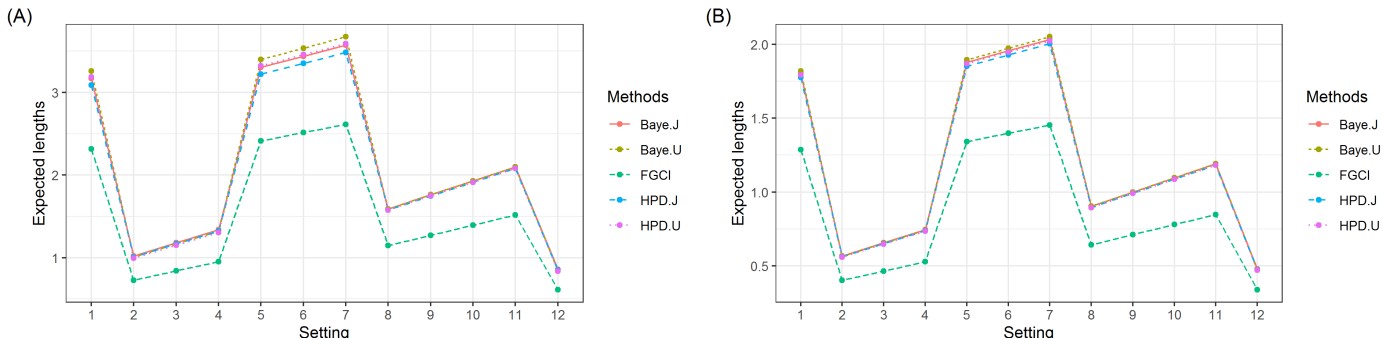

**Figure 6.** Comparison of expected lengths for estimating the 95% confidence interval for the common mean of several ZIG distributions with the samples containing noise: (**A**) $n_i = 30_3$ and (**B**) $n_i = 100_3$.

**Table 4.** Robustness studies with small amount of random noise for $k = 3$.

| Setting | $n_i$ | FGCI | | Baye.J | | Baye.U | | HPD.J | | HPD.U | |
|---|---|---|---|---|---|---|---|---|---|---|---|
| | | CP | EL | CP | EL | CP | EL | CP | EL | CL | EL |
| 1 | $30_3$ | 0.9416 | 2.3163 | 0.9629 | 3.1689 | 0.9453 | 3.2590 | 0.9734 | 3.0898 | 0.9593 | 3.1857 |
| | $100_3$ | 0.9360 | 1.2865 | 0.9589 | 1.8008 | 0.9447 | 1.8173 | 0.9658 | 1.7759 | 0.9531 | 1.7927 |
| 2 | $30_3$ | 0.9438 | 2.4155 | 0.9655 | 3.3055 | 0.9489 | 3.3999 | 0.9770 | 3.2233 | 0.9620 | 3.3233 |
| | $100_3$ | 0.9483 | 1.3416 | 0.9642 | 1.8774 | 0.9508 | 1.8948 | 0.9714 | 1.8516 | 0.9593 | 1.8690 |
| 3 | $30_3$ | 0.9509 | 2.5143 | 0.9683 | 3.4400 | 0.9515 | 3.5381 | 0.9772 | 3.3542 | 0.9650 | 3.4582 |
| | $100_3$ | 0.9562 | 1.3972 | 0.9695 | 1.9556 | 0.9573 | 1.9735 | 0.9747 | 1.9288 | 0.9655 | 1.9466 |
| 4 | $30_3$ | 0.9542 | 2.6142 | 0.9737 | 3.5729 | 0.9561 | 3.6748 | 0.9827 | 3.4841 | 0.9694 | 3.5919 |
| | $100_3$ | 0.9618 | 1.4517 | 0.9743 | 2.0316 | 0.9642 | 2.0504 | 0.9784 | 2.0036 | 0.9714 | 2.0225 |
| 5 | $30_3$ | 0.9467 | 1.1493 | 0.9775 | 1.5890 | 0.9643 | 1.5895 | 0.9754 | 1.5753 | 0.9638 | 1.5759 |
| | $100_3$ | 0.9195 | 0.6421 | 0.9636 | 0.9028 | 0.9516 | 0.9030 | 0.9605 | 0.8948 | 0.9490 | 0.8950 |
| 6 | $30_3$ | 0.9588 | 1.2716 | 0.9845 | 1.7642 | 0.9779 | 1.7648 | 0.9841 | 1.7490 | 0.9745 | 1.7496 |
| | $100_3$ | 0.9498 | 0.7110 | 0.9754 | 1.0000 | 0.9666 | 1.0001 | 0.9737 | 0.9911 | 0.9645 | 0.9913 |
| 7 | $30_3$ | 0.9707 | 1.3945 | 0.9894 | 1.9294 | 0.9837 | 1.9303 | 0.9893 | 1.9129 | 0.9827 | 1.9136 |
| | $100_3$ | 0.9653 | 0.7794 | 0.9827 | 1.0957 | 0.9768 | 1.0958 | 0.9820 | 1.0860 | 0.9745 | 1.0861 |
| 8 | $30_3$ | 0.9787 | 1.5159 | 0.9908 | 2.1003 | 0.9856 | 2.1011 | 0.9906 | 2.0821 | 0.9853 | 2.0830 |
| | $100_3$ | 0.9774 | 0.8479 | 0.9892 | 1.1914 | 0.9850 | 1.1916 | 0.9885 | 1.1809 | 0.9824 | 1.1810 |
| 9 | $30_3$ | 0.9105 | 0.6137 | 0.9641 | 0.8604 | 0.9518 | 0.8491 | 0.9566 | 0.8493 | 0.9417 | 0.8375 |
| | $100_3$ | 0.8814 | 0.3394 | 0.9428 | 0.4789 | 0.9313 | 0.4768 | 0.9342 | 0.4739 | 0.9204 | 0.4718 |
| 10 | $30_3$ | 0.9422 | 0.7267 | 0.9824 | 1.0216 | 0.9733 | 1.0081 | 0.9776 | 1.0085 | 0.9642 | 0.9944 |
| | $100_3$ | 0.9272 | 0.4020 | 0.9664 | 0.5676 | 0.9573 | 0.5651 | 0.9603 | 0.5617 | 0.9498 | 0.5592 |
| 11 | $30_3$ | 0.9622 | 0.8399 | 0.9891 | 1.1831 | 0.9828 | 1.1678 | 0.9853 | 1.1680 | 0.9775 | 1.1519 |
| | $100_3$ | 0.9560 | 0.4651 | 0.9835 | 0.6573 | 0.9780 | 0.6543 | 0.9800 | 0.6504 | 0.9726 | 0.6475 |
| 12 | $30_3$ | 0.9720 | 0.9513 | 0.9953 | 1.3403 | 0.9895 | 1.3231 | 0.9923 | 1.3234 | 0.9856 | 1.3051 |
| | $100_3$ | 0.9745 | 0.5278 | 0.9912 | 0.7455 | 0.9879 | 0.7422 | 0.9882 | 0.7377 | 0.9833 | 0.7343 |

## 4. Empirical Application of the Confidence Interval Estimation Methods with Real Data

Daily rainfall data supplied by the Upper Northern Region Irrigation Hydrology Center [21] were from the Chomthong, Mae Taeng, and Doi Saket districts in Chiang Mai, Thailand during September 2020 and 2021. Table 5 includes daily rainfall data from the three areas, and Figures 7 and 8 present histogram plots of the rainfall observations and Q-Q plots of the positive rainfall data following gamma distributions, respectively. We focused on estimating the daily rainfall data in these areas by applying the estimation methods for the confidence interval for the common mean of three ZIG distributions. By separating the rainfall data into non-zero and zero observations, it was possible to determine the best-fitting distribution for the rainfall data with positive values only. The lowest Akaike information criterion (AIC) and Bayesian information criterion (BIC) values in Tables 6 and 7, respectively, confirm that the gamma distribution was the best fit for all three non-zero rainfall datasets.

**Table 5.** The daily rainfall data from Chomthong, Mae Taeng, and Doi Saket in Chiang Mai, Thailand.

| Area | Daily Rainfall (mm) | | | | | | | | | |
|------|------|------|------|------|------|------|------|------|------|------|
| Chomthong | 2.3 | 0.0 | 0.0 | 0.0 | 0.0 | 3.5 | 0.0 | 14.2 | 0.0 | 0.0 |
| | 0.0 | 3.1 | 0.0 | 0.0 | 0.0 | 0.0 | 0.0 | 0.0 | 49.5 | 28.7 |
| | 4.0 | 0.0 | 4.8 | 0.0 | 0.0 | 0.07 | 0.0 | 4.5 | 12.2 | 0.0 |
| | 0.0 | 0.0 | 0.6 | 0.0 | 0.2 | 5.8 | 9.4 | 33.5 | 7.5 | 21.2 |
| | 4.0 | 0.0 | 23.2 | 2.2 | 12.6 | 33.8 | 10.2 | 0.0 | 0.0 | 0.2 |
| | 23.5 | 43.4 | 13.4 | 25.5 | 20.0 | 26.7 | 6.5 | 0.0 | 0.0 | 0.0 |
| Mae Taeng | 0.9 | 0.0 | 0.0 | 0.0 | 0.0 | 0.0 | 0.1 | 9.8 | 0.0 | 0.0 |
| | 0.0 | 0.0 | 0.0 | 0.3 | 0.0 | 0.0 | 2.3 | 0.3 | 14.3 | 22.0 |
| | 0.5 | 0.0 | 33.3 | 5.3 | 0.0 | 0.0 | 0.0 | 0.0 | 4.7 | 12.3 |
| | 0.0 | 0.1 | 8.0 | 0.0 | 0.0 | 6.7 | 1.2 | 4.7 | 39.0 | 19.5 |
| | 0.0 | 0.0 | 21.6 | 3.7 | 0.7 | 37.7 | 0.0 | 0.0 | 0.0 | 1.3 |
| | 15.7 | 0.0 | 5.1 | 0.0 | 13.5 | 4.3 | 15.6 | 0.0 | 0.0 | 0.0 |
| Doi Saket | 11.5 | 0.0 | 0.0 | 0.0 | 0.0 | 0.0 | 0.0 | 3.6 | 3.2 | 0.0 |
| | 0.0 | 0.0 | 0.0 | 0.0 | 0.0 | 0.0 | 0.0 | 0.0 | 30.1 | 19.6 |
| | 30.2 | 0.0 | 36.6 | 0.0 | 0.0 | 0.0 | 0.0 | 0.0 | 3.0 | 31.0 |
| | 0.0 | 30.0 | 54.4 | 0.0 | 1.1 | 3.0 | 15.5 | 14.2 | 42.8 | 0.5 |
| | 0.0 | 0.0 | 32.8 | 17.6 | 3.5 | 13.9 | 0.0 | 0.0 | 0.0 | 0.2 |
| | 16.2 | 7.6 | 0.9 | 1.2 | 20.5 | 11.6 | 1.2 | 0.0 | 0.0 | 0.0 |

**Table 6.** AIC values for fitting the positive rainfall data from Chomthong, Mae Taeng, and Doi Saket in Chiang Mai, Thailand.

| Area | Distribution | | | |
|------|-------|--------|-----------|--------|
| | **Gamma** | **Cauchy** | **Lognormal** | **Normal** |
| Chomthong | 231.7862 | 259.7445 | 239.6908 | 251.4082 |
| Mae Taeng | 198.9289 | 233.7925 | 204.2244 | 233.2959 |
| Doi Saket | 221.1700 | 252.2527 | 226.9918 | 241.2362 |

**Table 7.** BIC values for fitting the positive rainfall data from Chomthong, Mae Taeng, and Doi Saket in Chiang Mai, Thailand.

| Area | Distribution | | | |
|------|-------|--------|-----------|--------|
| | **Gamma** | **Cauchy** | **Lognormal** | **Normal** |
| Chomthong | 234.6542 | 262.6125 | 242.5587 | 254.2761 |
| Mae Taeng | 201.7313 | 236.5949 | 207.0268 | 236.0983 |
| Doi Saket | 223.9046 | 254.9873 | 229.7264 | 243.9708 |

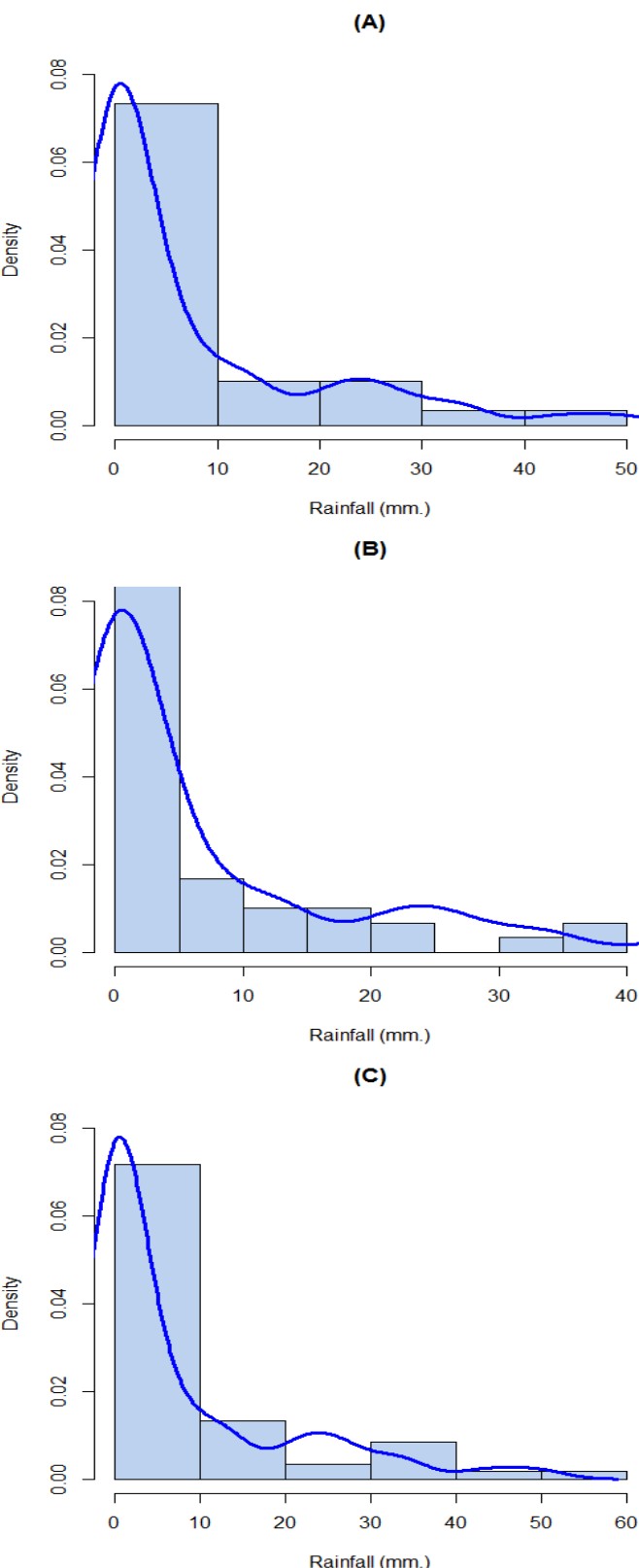

**Figure 7.** The densities of the rainfall datasets from Chiang Mai, Thailand: (**A**) Chomthong, (**B**) Mae Taeng, and (**C**) Doi Saket.

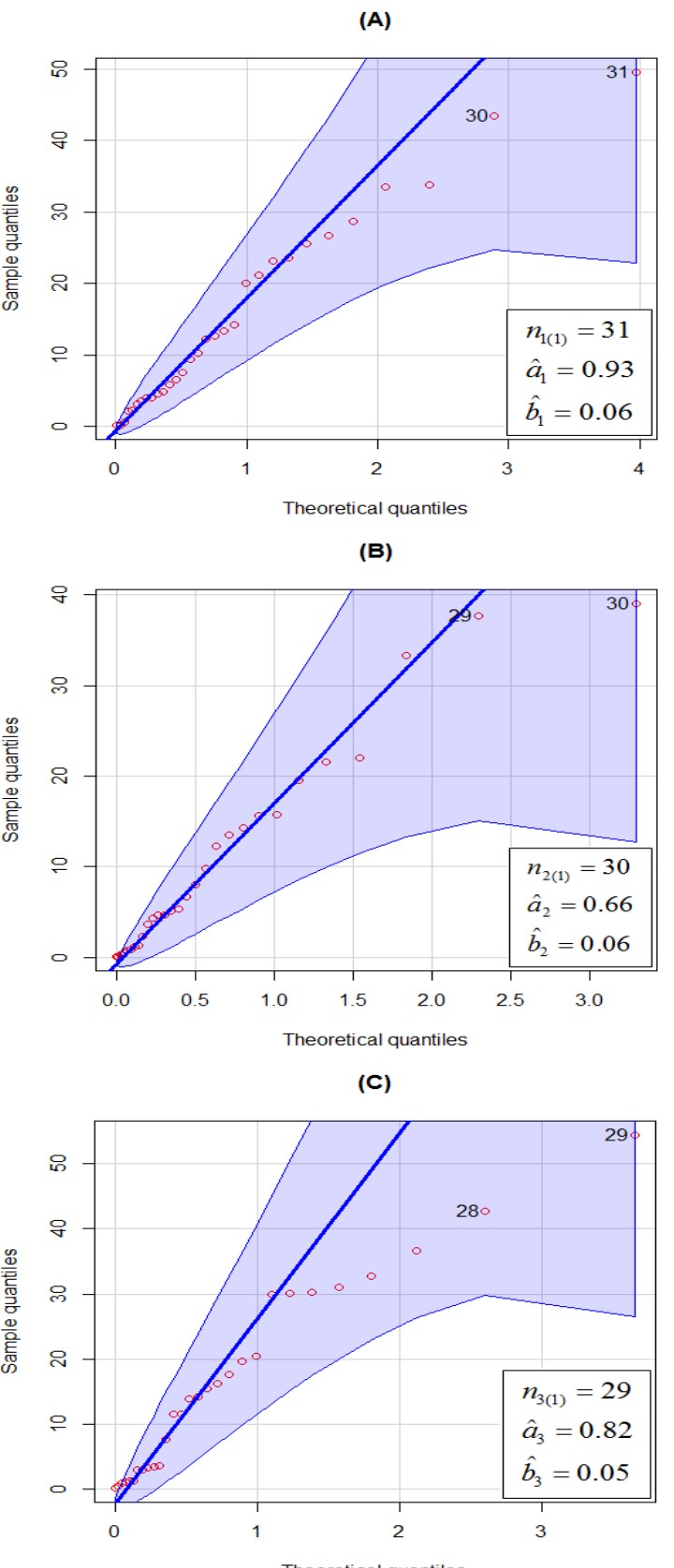

**Figure 8.** Q-Q plots of the non-zero observations in the daily rainfall datasets from Chiang Mai, Thailand: (**A**) Chomthong, (**B**) Mae Taeng, and (**C**) Doi Saket.

The summary statistics computed for the rainfall datasets from Chomthong, Mae Taeng, and Doi Saket in Chiang Mai, Thailand are reported in Table 8. The estimated confidence interval for the common mean of the three rainfall datasets was 5.79 mm/day. Table 9 summarizes the computed 95% confidence interval for the common mean for the three rainfall datasets using the proposed methods. The length of the confidence interval estimated via fiducial GCI was the shortest, which supports the simulation results for $k = 3$ in the previous section. Thus, we recommend the fiducial GCI for estimating the confidence interval for the common mean of several ZIG distributions.

**Table 8.** Parameter estimates for the three rainfall datasets.

| Area | Parameter Estimates | | | | | | |
|------|------|------|------|------|------|------|------|
| | $n_i$ | $\hat{\delta}_i$ | $\hat{a}_i$ | $\hat{b}_i$ | $\hat{\mu}_i$ | $\hat{\sigma}_i^2$ | $\hat{\eta}_i$ |
| Chomthong | 60 | 0.48 | 0.93 | 0.06 | 2.17 | 0.69 | 7.50 |
| Mae Taeng | 60 | 0.50 | 0.66 | 0.06 | 1.83 | 0.76 | 5.07 |
| Doi Saket | 60 | 0.52 | 0.82 | 0.05 | 2.19 | 0.84 | 7.62 |

**Table 9.** The 95% confidence interval estimates for the common mean of the three rainfall datasets from Chiang Mai, Thailand.

| Methods | 95% Confidence Interval | | Lengths |
|---------|-------|-------|---------|
| | Lower | Upper | |
| FGCI | 3.3521 | 4.7108 | 1.3587 |
| Baye.J | 4.1330 | 6.6412 | 2.5082 |
| Baye.U | 4.0183 | 6.5264 | 2.5081 |
| HPD.J | 4.1837 | 6.6781 | 2.4944 |
| HPD.U | 4.0021 | 6.4931 | 2.4910 |

## 5. Discussion

Estimating the confidence interval for the common mean of several gamma distributions was first reported by Yan [5]. Meanwhile, Maneerat and Niwitpong [6] proposed estimation methods for the confidence interval for the common mean of several delta-lognormal distributions (a lognormal distribution with zero observations) using the fiducial GCI and HPD interval based on the Jeffreys rule prior. In this study, we extended these ideas to construct estimates for the confidence interval for the common mean of several ZIG distributions. Specifically, we proposed several approaches based on the fiducial GCI and Bayesian and HPD methods based on the Jeffreys rule or uniform priors. A coverage probability close to or greater than the nominal confidence level of 0.95 and the shortest expected length were used to select the best-performing confidence interval. The results indicate that, while the Bayesian and HPD coverage probability were close to or greater than the nominal confidence level of 0.95, those of the fiducial GCI were similarly close to or greater than that level, and their expected length was the shortest. However, the results from a comparative simulation study show that the coverage probabilities of the fiducial GCI, the Bayesian, and HPD interval based on Jeffreys rule or uniform prior were greater than or close to the nominal confidence level of 0.95 under most circumstances. As the sample sizes increased, the coverage probabilities of all of the proposed methods performed better but were still under the nominal confidence level of 0.95. When the sample sizes were increased, the expected lengths of all of the proposed methods became shorter, whereas when the shape parameter was increased, the expected lengths of all of the proposed methods became longer. When considering the expected lengths, those of the fiducial GCI were the shortest under most circumstances. If the proportion of zero values increased, the expected lengths of all of the proposed methods became shorter. However, the coverage probabilities of the fiducial GCI were lower than the nominal confidence level of 0.95 in some cases. The HPD interval based on the Jeffreys rule or uniform prior outperformed

the fiducial GCI. Overall, the fiducial GCI and the HPD interval based on the Jeffreys rule or uniform prior performed the best in the simulation study because they fulfilled the requirements for both criteria. Although Kaewprasert et al. [1] claimed that Bayesian and HPD methods are the most effective for estimating the mean and the difference between the means of ZIG distributions, our findings for the data and scenario used in this study contradict their claims because the range of intervals for the common mean was wider than when using the Bayesian and HPD methods. According to our results, the fiducial GCI consistently supplied the smallest expected length and a suitable coverage probability for both $k = 3$ and $k = 5$. However, in certain instances, the HPD based on Jeffreys rule prior produced results that were consistent with those of Kaewpraset et al. [1].

In addition, we calculated the confidence interval for the common mean of three rainfall datasets from Chiang Mai, Thailand using the proposed methods. We found that the fiducial GCI once again performed the best in this empirical scenario. Our approach may be useful for estimating the rainfall in September, as this information could be important for residents in the hilly and forested regions of places such as Chiang Mai who want to avoid flooding and landslides.

## 6. Conclusions

We constructed estimators for the confidence interval for the common mean of several ZIG distributions using the fiducial GCI and Bayesian and HPD methods based on the Jeffreys rule or uniform prior. The coverage probability and expected length were used to assess how well they performed in various scenarios. According to the findings from the simulation study, the coverage probabilities of the fiducial GCI were greater than the nominal confidence level of 0.95, and its expected lengths were the shortest in almost all cases for $k = 3$ and $k = 5$. The efficacies of the proposed methods were tested using real daily rainfall datasets from Chomthong, Mae Taeng, and Doi Saket in Chiang Mai, Thailand. Once again, the fiducial GCI outperformed the other methods by providing the shortest length of the confidence interval, which is the same as the simulation study results. Therefore, the fiducial GCI is recommended for estimating the confidence interval for the common mean of several ZIG distributions, while the HPD based on the Jeffreys rule or uniform prior could also be used in some scenarios.

**Author Contributions:** Conceptualization, S.N.; methodology, S.-A.N. and S.N.; software, T.K.; validation, T.K., S.-A.N. and S.N.; formal analysis, T.K. and S.N.; investigation, S.-A.N. and S.N.; resources, S.-A.N.; data curation, T.K.; writing—original draft preparation, T.K.; writing—review and editing, S.-A.N. and S.N.; visualization, S.-A.N.; supervision, S.-A.N. and S.N.; project administration, S.-A.N.; funding acquisition, S.N. All authors have read and agreed to the published version of the manuscript.

**Funding:** This research has received funding support from the National Science, Research and Innovation Fund (NSRF), and King Mongkut's University of Technology North Bangkok: KMUTNB-FF-66-03.

**Institutional Review Board Statement:** Not applicable.

**Informed Consent Statement:** Not applicable.

**Data Availability Statement:** The real datasets of rainfall were obtained from the Upper Northern Region Irrigation Hydrology Center [21].

**Acknowledgments:** The first author wishes to express gratitude for financial support provided by the Thailand Science Achievement Scholarship (SAST).

**Conflicts of Interest:** The authors declare no conflict of interest.

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
