# Peer review of "Confidence Interval Estimation for the Common Mean of Several Zero-Inflated Gamma Distributions"

_symmetry, doi:10.3390/sym15010067_

Round 1

Reviewer 1 Report

This article proposed confidence intervals based on fiducial and Bayesian inference for the common mean of several zero-inflated gamma distribution.  The whole work is nicely presented. However, I have a few comments as follows.

1.  For the simulation study section, we can see that fiducial method performs better than other methods according to coverage probabilities and average interval lengths. However, I am wondering if the fiducial method is well balanced. Please provide some information about the tail error rates.

2. The simulation results would be more convincing if the authors could illustrate the robustness of the proposed methods.

3. The quality of Figure 5 and Figure 6 can be further improved. In addition, some statistical values (e.g. estimates for parameters)should be presented in Figure 6 to indicate the fit is appropriate. 

Author Response

 Dear Reviewers,
We are grateful for the reviewer’s valuable comments and have all suggestions
seriously. Reviewer’s critiques addressed section by section in this document, and
corrections were incorporated in manuscript accordingly
. Please see attached.

Author Response

(The authors gave the same response as above.)

Reviewer 3 Report

sorry, I do not understand the core content of this article

Author Response

 Dear Reviewers,
We are grateful for the reviewer’s valuable comments.

Round 2

Reviewer 1 Report

It seems that all the methods proposed in this paper are unbalanced. There is still space for further improvement. However, I think the current version can be accepted even if the tail error rates can not be fixed.

Reviewer 2 Report

All my comments have been taken into account and the revised version of the manuscript is suitable for publication in Symmetry.